# A new perspective on the nature of dropout

## Abstract

In this work, we explore the average behavior of the learning process with dropout in the contexts of linear regression, generalized linear models, matrix factorization, and fully-connected neural networks with dropout in the last layer. Initially, we find that the average behavior does not distinguish the original dropped-out quantity. The implication of this is that the dropout-induced regularization and optimization are ambiguous from the perspective of the average behavior. To resolve this, we reformulate the average behavior based on the elementary operations that a practitioner is able to apply in the learning process with dropout. Then, we disambiguate the dropout-induced regularization and optimization from the perspective of each reformulation. In the context of linear regression, we show that all of the reformulations result in the same predictions at test time, where the invariant in these predictions is the square of the coefficient of variation of the dropout distribution. More broadly, we demonstrate that the penalty term under dropout depends on the data, parameters, and predictions at train time, when the mean of the dropout distribution is not equal to one.

## 1 Introduction

Dropout is a multiplicative method that introduces randomness into the learning process in order to improve generalization performance. Over time, researchers have analyzed the average behavior of the learning process with dropout in several contexts of machine learning, where they established connections between the average behavior and methods of regularization, such as Tikhonov regularization and ridge regression (Tikhonov, 1963; Hoerl & Kennard, 1970; Webb, 1994; Bishop, 1995; Srivastava, 2013; Wager et al., 2013; Baldi & Sadowski, 2013; Srivastava et al., 2014; Helmbold & Long, 2015; Cavazza et al., 2018; Mianjy et al., 2018; Helmbold & Long, 2018; Wei et al., 2020; Arora et al., 2021).

To do so, they constructed a multiplicative mask based on a random sample from a Bernoulli distribution. Then, they formed the loss under dropout and decomposed its expectation into a sum of a loss term and a regularization term. From there, they employed the decomposition in order to interpret the dropout-induced regularization and optimization.

More recently, the work of Clara et al. (2024) juxtaposed the average behavior and gradient descent iterates of linear regression with dropout in order to analyze the dropout-induced regularization and optimization. Moreover, they decomposed the expected loss under dropout based on a random sample from an abstract probability distribution in the discussion and outlook section of the work.

Relatedly, we explore the average behavior of the learning process with dropout in the contexts of linear regression, generalized linear models, matrix factorization, and fully-connected neural networks with dropout in the last layer.

- To this end, we construct each multiplicative mask based on a random sample from a probability distribution that has finite but non-zero mean and finite variance.
- Then, we show that each decomposition is many-to-one in terms of the dropped-out quantity.

In other words, we find that the average behavior does not distinguish the original dropped-out quantity in these contexts of machine learning.

The consequence of this property of the average behavior is that our interpretation of the dropout-induced regularization and optimization is ambiguous, when we think from the perspective of these decompositions. That is, we know that these decompositions contain a scalar, which we are able to leave intact or attribute to the fit and predict variables in accord with the rules of algebra. But, we do not know how to identify the original implementation details in the mathematics of these decompositions due to the many-to-one property. Thus, if we restrict ourselves to algebraic reason, then we conclude that our interpretations are arbitrary.

To illustrate our point, we recall the work of Srivastava et al. (2014) on the average behavior of linear regression with dropout.

- There, they dropped out the design matrix, minimized the expected loss under dropout with respect to the original parameters, and chose to leave the scalar intact in the initial decomposition. Subsequently, they posited a bounded interpretation of the regularization strength.

- Then, they chose to attribute the scale to the parameters in an equivalent decomposition, which led them to posit an unbounded interpretation of the regularization strength.

In this way, we observe that the work of Srivastava et al. (2014) produced two incompatible interpretations of the regularization strength in the same optimization problem, where the re-scaled parameters were an algebraic artifact in the optimization problem of interest. Moreover, we contend that such ambiguities went undetected in the other contexts of machine learning, since the researchers chose to focus on decompositions in which the scalar is one, i.e., the literature converged on the programmatic convention of inverted dropout (Wager et al., 2013; Baldi & Sadowski, 2013; Helmbold & Long, 2015; Cavazza et al., 2018; Mianjy et al., 2018; Helmbold & Long, 2018; Wei et al., 2020; Arora et al., 2021).

To resolve this, we introduce a principle of machine learned interpretation, which goes beyond algebraic reason. Namely, we enumerate the ways in which a practitioner is able to apply elementary operations in the learning process with dropout.

- For the non-matrix factorization contexts, we include the choices of column normalization and which quantity to scale.

- Otherwise, we include the choice of which quantity to scale.

Correspondingly, we map the original fit and predict variables into fit and predict variables, which model the elementary operations.

- Next, we reformulate each decomposition in terms of the operational fit and predict variables.

- Then, we disambiguate the dropout-induced regularization and optimization from the perspective of each reformulation.

The result of this is that we produce interpretations that are bespoke to each elementary experimental setup.

In the context of linear regression, we show that all of the reformulations result in the same predictions at test time, where the invariant in these predictions is the square of the coefficient of variation of the dropout distribution; see Figure 1. For the other contexts, we do not possess a closed-form solution, so we leave open the question of whether or not the reformulations all result in the same output at test time.

Beyond this, we discover that the difference between the expected loss under dropout and the original loss term of interest depends on the data, parameters, and predictions at train time, when the mean of the dropout distribution is not equal to one. Historically, the literature had not explored this generalization, which led to the belief that the penalty term under dropout depends on neither the entire data nor the predictions at train time in the contexts of linear regression, generalized linear models, and matrix factorization (Wager et al., 2013; Helmbold & Long, 2015; 2018).

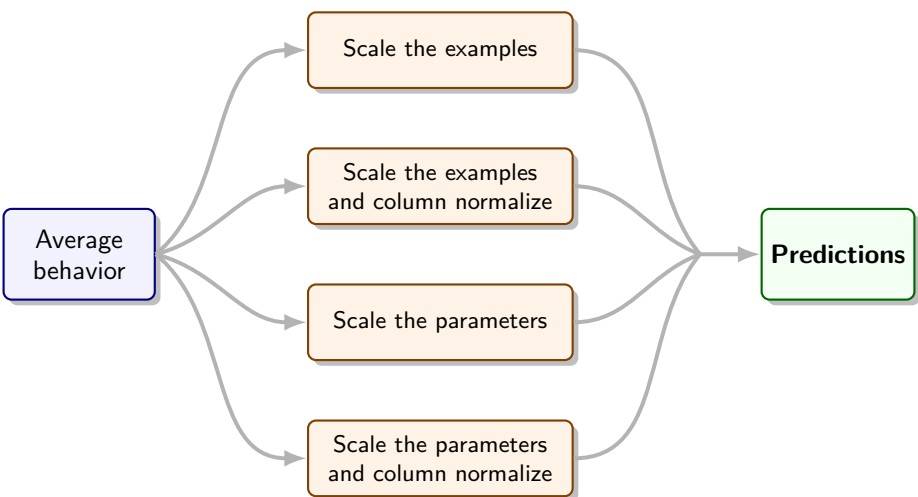

Figure 1: We reformulate the average behavior of linear regression with dropout in four different ways in order to disambiguate the dropout-induced regularization and optimization. Then, we show that all of the reformulations result in the same predictions at test time, where the invariant in these predictions is the square of the coefficient of variation of the dropout distribution.

## 2  Linear Regression

We begin with the problem of linear regression, since this enables us to establish notation and showcase our approach in a tractable setting. Here,

$$J(\theta) = \left|\left|Y - X\theta\right|\right|_2^2 \tag{1}$$

denotes the squared error loss function. Specifically, the term $Y$ is the collection of $m$ targets, the term $X$ is the design matrix with $m$ examples and $n$ features, and the term $\theta$ is the collection of $n$ parameters. Overall, the objective is to find an element of the set $\arg\min_\theta J(\theta)$, which corresponds to a collection of parameters that minimize the loss function of interest.

Correspondingly,

$$J(\theta) = \left|\left|Y - X\theta\right|\right|_2^2 + \left|\left|\Gamma\theta\right|\right|_2^2, \tag{2}$$

denotes the squared error loss function with a Tikhonov regularization term, where $\Gamma$ is the Tikhonov matrix, which may, in general, be rectangular. [1] Moreover, if the Tikhonov matrix is a scalar multiple $\sqrt{\lambda}$ of a matrix $G$, i.e., $\Gamma = \sqrt{\lambda}G$, then we call $\lambda$ the regularization strength. Furthermore, if $G = \mathbb{I}$ is the identity matrix, then we call the problem ridge regression

$$J(\theta) = \left|\left|Y - X\theta\right|\right|_2^2 + \lambda\left|\left|\theta\right|\right|_2^2,$$

where $\lambda$ corresponds to the so-called ridge parameter.

The standard penalty term is the difference between Eq. 2 and Eq. 1, i.e., the Tikhonov regularization term. In general, we define a penalty term to be the difference between a regularized loss term and the original loss term of interest, which corresponds to Eq. 1 in the context of linear regression.

### 2.1  Linear regression with dropout

To study linear regression with dropout, we draw independent and identically distributed random variables $d_j \overset{\text{iid}}{\sim} \mathfrak{D}$ from a probability distribution that has finite but non-zero mean $\mathbb{E}[d_j] = \mu$ and finite variance $\text{Var}(d_j) = \sigma^2$, where $j = 1, \ldots, n$. [2] Next, we construct a dropout mask $\delta = \text{diag}(d_1, \ldots, d_n)$, which

---

[1]Throughout this work, we do not use a subscript to distinguish the loss function.

[2]We make the non-zero mean assumption such that we are able to form the coefficient of variation, i.e., $\frac{\sigma}{\mu}$.

is a diagonal matrix with the dropout random variables on the diagonal. Then, we form the composite dropped-out product

$$X\delta\theta = (X\delta)\theta = X(\delta\theta),$$

where $X\delta$ and $\delta\theta$ correspond to the dropped-out design matrix and parameters, respectively.

### 2.1.1 Expected loss under dropout

Accordingly, we form the squared error loss under dropout

$$J(\theta) = \left|\left|Y - X\delta\theta\right|\right|_2^2.$$

Next, we consider the expected loss under dropout

$$\mathbb{E}[J(\theta)] = \mathbb{E}[\left|\left|Y - X\delta\theta\right|\right|_2^2], \tag{3}$$

since this expectation value enables us to analyze the average behavior of linear regression with dropout. [3] Then, we have that Eq. 3 decomposes into a sum of a squared error loss term and a data-dependent Tikhonov regularization term

$$\mathbb{E}[J(\theta)] = \left|\left|Y - \mu X\theta\right|\right|_2^2 + \left|\left|\Gamma\theta\right|\right|_2^2, \tag{4}$$

where the Tikhonov matrix is a scalar multiple $\sigma$ of a diagonal matrix G such that

$$(G)_{jj} = \text{diag}\left(||X_1||_2, \ldots, ||X_n||_2\right)_{jj} = \left|\left|X_j\right|\right|_2,$$

$X_j$ denotes the $j$th column of the design matrix, and $j = 1, \ldots, n$; see the details in section A.1 of the appendix.

Notably, we observe that Eq. 4 is many-to-one in terms of the original dropped-out quantity due to the composite nature of dropout.

- This implies that the average behavior of linear regression with dropout does not distinguish the original dropped-out quantity.

- The consequence of this property of the average behavior is that our interpretation of the dropout-induced regularization and optimization is ambiguous, when we think from the perspective of Eq. 4.

Concretely, we recall the work of Srivastava et al. (2014). [4] To do so, we substitute $\mu = p$ and $\sigma = \sqrt{p(1-p)}$ in Eq. 4, where $p \in (0, 1)$ denotes the probability of success in the context of a Bernoulli distribution. Also, we fix the optimization problem of interest in terms of the original parameters.

- Initially, we observe that the regularization strength appears to equal the variance $\lambda = \sigma^2 = p(1-p)$, which is bounded from above by $\frac{1}{4}$.

- Next, we attribute the scale to the parameters and create the re-scaled parameters $\tilde{\theta} = \mu\theta = p\theta$. Then, we observe that the regularization strength appears to equal the odds against $\lambda = (\frac{\sigma}{\mu})^2 = \frac{1-p}{p}$, which is unbounded from above.

Therefore, we find that there exist two incompatible interpretations of the regularization strength, when we think from the perspective of Eq. 4 and follow the algebraic reasons in the work of Srivastava et al. (2014).

---

[3]Based on the construction in section 2.1, our approach only requires one equation, whereas the conventional approach requires two separate equations due to the nature of the Hadamard product.

[4]In particular, we focus on section 9.1 in the work of Srivastava et al. (2014), which stems from section 4.1 in the master's thesis of Srivastava (2013).

## 2.2 Reformulations of the decomposition

From the lessons in section 2.1.1, we know that Eq. 4 requires us to develop a principle of interpretation that goes beyond algebraic reason in order to interpret the dropout-induced regularization and optimization. To this end, we enumerate the ways in which a practitioner is able to apply elementary operations in linear regression with dropout in sections 2.2.1 and 2.2.2.

Correspondingly, we map the original fit and predict variables into fit and predict variables, which model the elementary operations. Next, we reformulate the decomposition in terms of the operational fit and predict variables. Then, we disambiguate the dropout-induced regularization and optimization and generate predictions at test time from the perspective of each reformulation.

### 2.2.1 Scale the examples

To carry this out, we begin with a practitioner, who chooses to scale the examples in accord with standard dropout. Next, we map the original design matrix to the re-scaled design matrix $\tilde{X} = \mu X = \mathbb{E}[X\delta]$. Then, we have that the decomposition becomes

$$\mathbb{E}[J(\theta)] = \left|\left|Y - \tilde{X}\theta\right|\right|_2^2 + (\frac{\sigma}{\mu})^2\left|\left|\tilde{G}\theta\right|\right|_2^2, \tag{5}$$

where $(\tilde{G})_{jj} = \left|\left|\mu X_j\right|\right|_2 = \left|\left|\tilde{X}_j\right|\right|_2$ for $j = 1, \ldots, n$.

Accordingly, we find that the regularization strength equals the square of the coefficient of variation $\lambda = (\frac{\sigma}{\mu})^2$, when we think about the nature of dropout from the perspective of Eq. 5. Moreover, we observe that the optimization problem of interest is in terms of the original parameters and there is symmetry in the appearance of the re-scaled design matrix in the loss term and regularization term.

Next, we consider the generation of a prediction at test time. To do so, we compute the normal equations in order to solve the optimization problem of interest

$$(\tilde{X}^t\tilde{X} + (\frac{\sigma}{\mu})^2\tilde{G}^t\tilde{G})\theta = \tilde{X}^tY,$$

where the solution is

$$\theta = (\tilde{X}^t\tilde{X} + (\frac{\sigma}{\mu})^2\tilde{G}^t\tilde{G})^{-1}\tilde{X}^tY,$$

when we are able to perform the inverse; see the details in section A.1.2 of the appendix.

Subsequently, we map a test example to a re-scaled test example $\tilde{x} = \mu x$ in order to form a prediction

$$\tilde{x} \cdot \theta = \tilde{x} \cdot (\tilde{X}^t\tilde{X} + (\frac{\sigma}{\mu})^2\tilde{G}^t\tilde{G})^{-1}\tilde{X}^tY. \tag{6}$$

Then, we have that the prediction is equivalent to a prediction from a particular form of linear regression with Tikhonov regularization

$$\mu x \cdot \frac{1}{\mu}(X^tX + (\frac{\sigma}{\mu})^2G^tG)^{-1}X^tY = x \cdot (X^tX + (\frac{\sigma}{\mu})^2G^tG)^{-1}X^tY.$$

Similarly, we consider a practitioner, who chooses to scale the examples and column normalize the design matrix such that $||X_j||_2 = 1$ for $j = 1, \ldots, n$. Then, we have that the decomposition becomes

$$\mathbb{E}[J(\theta)] = \left|\left|Y - \tilde{X}\theta\right|\right|_2^2 + \sigma^2\left|\left|\theta\right|\right|_2^2. \tag{7}$$

Consequently, we find that the regularization strength equals the variance $\lambda = \sigma^2$, when we think about the nature of dropout from the perspective of Eq. 7. Moreover, we observe that the optimization problem of interest is in terms of the original parameters and there is asymmetry in the appearance of the re-scaled design matrix in the loss term and regularization term.

Correspondingly, we consider the generation of a prediction at test time. To do so, we compute the normal equations, obtain the solution, and form the prediction

$$\tilde{x} \cdot \theta = \tilde{x} \cdot (\tilde{X}^t \tilde{X} + \sigma^2 \mathbb{I})^{-1} \tilde{X}^t Y. \tag{8}$$

Then, we have that the prediction is equivalent to a prediction from a particular form of ridge regression

$$\mu x \cdot \frac{1}{\mu}(X^t X + (\frac{\sigma}{\mu})^2 \mathbb{I})^{-1} X^t Y = x \cdot (X^t X + (\frac{\sigma}{\mu})^2 \mathbb{I})^{-1} X^t Y,$$

where the ridge parameter equals the square of the coefficient of variation. [5]

### 2.2.2 Scale the parameters

Ensuingly, we consider a practitioner, who chooses to scale the parameters. Next, we map the original parameters to the re-scaled parameters $\tilde{\theta} = \mu\theta = \mathbb{E}[\delta\theta]$. Then, we have that the decomposition becomes

$$J(\tilde{\theta}) = \left|\left| Y - X\tilde{\theta} \right|\right|_2^2 + (\frac{\sigma}{\mu})^2 \left|\left| G\tilde{\theta} \right|\right|_2^2, \tag{9}$$

which is a squared error loss function of the re-scaled parameters with a Tikhonov regularization term.

Accordingly, we find that the regularization strength equals the square of the coefficient of variation $\lambda = (\frac{\sigma}{\mu})^2$, when we think about the nature of dropout from the perspective of Eq. 9. Moreover, we observe that the optimization problem of interest is in terms of the re-scaled parameters and there is symmetry in the appearance of the re-scaled parameters in the loss term and regularization term.

Next, we consider the generation of a prediction at test time. To do so, we compute the normal equations in order to solve the optimization problem of interest

$$(X^t X + (\frac{\sigma}{\mu})^2 G^t G)\tilde{\theta} = X^t Y,$$

where the solution is

$$\tilde{\theta} = (X^t X + (\frac{\sigma}{\mu})^2 G^t G)^{-1} X^t Y,$$

when we are able to perform the inverse; see the details in section A.1.3 of the appendix.

Subsequently, we form a prediction

$$x \cdot \tilde{\theta} = x \cdot (X^t X + (\frac{\sigma}{\mu})^2 G^t G)^{-1} X^t Y, \tag{10}$$

which does not require us to attribute the scale at test time. Then, we observe that the prediction is equivalent to the prediction in Eq. 6, which shows that the two interpretations result in the same predictions at test time.

Similarly, we consider a practitioner, who chooses to scale the parameters and column normalize the design matrix. Then, we have that the decomposition becomes

$$J(\tilde{\theta}) = \left|\left| Y - X\tilde{\theta} \right|\right|_2^2 + (\frac{\sigma}{\mu})^2 \left|\left| \tilde{\theta} \right|\right|_2^2, \tag{11}$$

which is a ridge regression problem in terms of the re-scaled parameters.

Consequently, we find that the regularization strength equals the square of the coefficient of variation $\lambda = (\frac{\sigma}{\mu})^2$, when we think about the nature of dropout from the perspective of Eq. 11. Moreover, we observe that

---

[5]From a Bayesian point of view, we recognize that the square of the coefficient of variation in the prediction at test time $(\frac{\sigma}{\mu})^2 = \frac{\sigma^2}{\mu^2}$ corresponds to the ridge parameter in a normal-normal model such that $\mu^2$ is the scalar multiple of the identity in the prior distribution case and $\sigma^2$ is the scalar multiple of the identity in the likelihood distribution case.

the optimization problem of interest is in terms of the re-scaled parameters and there is symmetry in the appearance of the re-scaled parameters in the loss term and regularization term.

Correspondingly, we consider the generation of a prediction at test time. To do so, we compute the normal equations, obtain the solution, and form the prediction

$$x \cdot \tilde{\theta} = x \cdot (X^t X + (\frac{\sigma}{\mu})^2 \mathbb{I})^{-1} X^t Y. \tag{12}$$

Then, we observe that the prediction is equivalent to the prediction in Eq. 8, which shows that the two interpretations result in the same predictions at test time.

## 2.3 Dual interpretations

From the reformulations of the decomposition in section 2.2, we found in section 2.2.1 and 2.2.2 that there exist multiple interpretations of the average behavior of linear regression with dropout such that each interpretation details the regularization strength and optimization problem to solve based on the ways in which a practitioner is able to scale a quantity and choose whether or not to column normalize the design matrix. Moreover, we demonstrated that these interpretations are dual in the sense that they form pairs, where each pair results in the same predictions at test time.

- That is, we showed that the interpretations from the perspectives of Eq. 5 and Eq. 9 result in predictions from a particular form of linear regression with Tikhonov regularization, when we reduce the predictions to a canonical form.

- Then, the interpretations from the perspectives of Eq. 7 and Eq. 11 result in predictions from a particular form of ridge regression, when we reduce the predictions to a canonical form.

Consequently, we know that the ways in which a practitioner is able to scale a quantity do not change the predictions at test time. However, the pairs of predictions leave open the question:

- Does the choice of whether or not to column normalize the design matrix change the predictions at test time?

### 2.3.1 Equivalence of predictions at test time

To answer this question, we employ more pedantic notation in order to show that a prediction from the particular form of linear regression with Tikhonov regularization is equivalent to a prediction from the particular form of ridge regression; see the details in section A.1.4 in the appendix.

- Thereupon, we know that the ways in which a practitioner is able to scale a quantity and choose whether or not to column normalize the design matrix do not change the predictions at test time.

Additionally, we observe that the invariant in these predictions is the square of the coefficient of variation $(\frac{\sigma}{\mu})^2$ of the dropout distribution.

Thus, we find that there are local and global hyper-parameters, when we think about the average behavior in an operational way. [6]

- Moreover, we observe that there is no need to re-scale at test time, when we follow the interpretations in section 2.2.2.

The reason is that we manage the scale through the change of the optimization problem of interest, where the fitted re-scaled parameters are fixed at test time.

---

[6]For example, if we consider Eq. 7, then we observe that the local regularization strength is equal to the variance. However, this is not always equal to the square of the coefficient of variation, which is the global hyper-parameter that we observe in the predictions at test time.

### 2.4 Penalty term under dropout

Correspondingly, we observe that the predictions resemble either the particular form of linear regression with Tikhonov regularization or the particular form of ridge regression, which leads us to ask:

- Why would a practitioner want to use dropout in place of standard regularization methods such as Tikhonov regularization or ridge regression?

To answer this question, we examine the penalty term under dropout and analyze the ways in which it differs from the standard penalty term. Namely, we begin with the penalty term under dropout

$$\mathbb{E}[J(\theta)] - J(\theta) = 2(1-\mu)\langle Y, X\theta \rangle_2 + (\mu^2 - 1)||X\theta||_2^2 + ||\Gamma\theta||_2^2, \tag{13}$$

which is the difference between the decomposition in Eq. 4 and the standard squared error loss function in Eq. 1.

From the right-hand side of Eq. 13, we have that the penalty term under dropout includes two regularization terms that go beyond the Tikhonov regularization term.

- In particular, the first term corresponds to a scalar multiple of the inner product between the targets and predictions at train time $2(1-\mu)\langle Y, X\theta \rangle_2$, which shows that the penalty term under dropout depends on the overlap between the targets and predictions at train time.

- Then, the second term corresponds to a scalar multiple of the square of the norm of the predictions at train time $(\mu^2 - 1)||X\theta||_2^2$, which shows that the penalty term under dropout depends on the size of the predictions at train time.

Interestingly, we observe that the two non-standard terms on the right-hand side of Eq. 13 vanish, when a practitioner picks a dropout distribution such that $\mu = 1$, i.e., inverted dropout or Gaussian dropout. Otherwise, the entire penalty term under dropout remains intact, when a practitioner picks a dropout distribution such that $\mu \neq 1$, i.e., a Bernoulli distribution with $\mu = p \in (0, 1)$.

#### 2.4.1 The zero mean additive case of dropout

In the case of $\mu = 1$, there is a correspondence between dropout and so-called zero mean additive noise, which is the connection that Wager, Wang, and Liang leveraged in the context of generalized linear models in order to extend the work of Tikhonov, Hoerl, Kennard, Webb, Bishop, and others on Tikhonov regularization and ridge regression (Tikhonov, 1963; Hoerl & Kennard, 1970; Webb, 1994; Bishop, 1995; Wager et al., 2013). Then, Helmbold and Long built off of this work and claimed that the penalty term under dropout only depends on the labeled examples in the context of deep learning (Helmbold & Long, 2015; 2018).

Relatedly, the Keras, MLX, PyTorch, and TensorFlow libraries only contain implementations of inverted dropout and Gaussian dropout, where $\mu = 1$ (Chollet et al., 2015; Abadi et al., 2016; Paszke et al., 2019; Hannun et al., 2023).

- This means that the penalty term under dropout collapses to the standard penalty term, when a practitioner chooses to implement linear regression with dropout based on these popular machine learning libraries.

Thus, there has been limited theoretical and experimental motivation to explore the average behavior of linear regression with dropout due to the historical fixation on the case of $\mu = 1$.

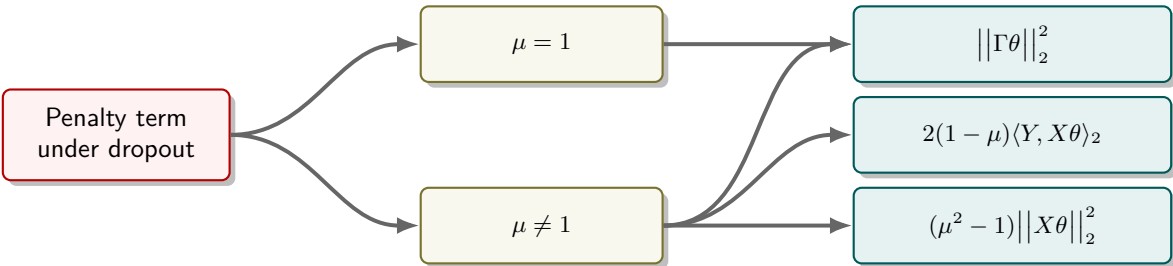

Figure 2: In the case of $\mu = 1$, the penalty term under dropout corresponds to Tikhonov regularization, where the Tikhonov matrix takes the form that we observe in Eq. 4. Otherwise, the penalty term under dropout generalizes this particular form of Tikhonov regularization with the addition of two non-standard terms, which depend on the overlap between the targets and predictions at train time as well as the size of the predictions at train time.

### 2.4.2 The non-zero mean additive case of dropout

However, we learned in section 2.3.1 that all of the interpretations result in the same predictions at test time, where the interpretations in sections 2.2.1 and 2.2.2 have a one-time and zero cost of re-scale, respectively. [7]

- Therefore, we find that the case of $\mu \neq 1$ requires at most a one-time cost of re-scale at test time, where the case differs from Tikhonov regularization and ridge regression in that the penalty term under dropout depends on the overlap between the targets and predictions at train time and the size of the predictions at train time.

In other words, the case of $\mu \neq 1$ reveals that the case of $\mu = 1$ is a restrictive convention that has obfuscated the average behavior of linear regression with dropout in theory and practice; see Figure 2.

## 3 Discussion and outlook

Now that we have examined the average behavior of linear regression with dropout, we turn our attention to the other contexts of machine learning that we mentioned in the introduction. To this end, we explore the average behavior of matrix factorization with dropout in section B and the average behavior of a fully-connected neural network with dropout in the last layer in section C.

In sections B.1.1 and C.1.1, we find that the decomposition of the expected loss under dropout is many-to-one in terms of the original dropped-out quantity. The implication of this is that the dropout-induced regularization and optimization are ambiguous, when we think from the perspective of these decompositions.

Consequently, we reformulate the decomposition in terms of the matrix factor to scale in section B.2, where we find that the regularization strength is always equal to the square of the coefficient of variation. Then, we reformulate the decomposition in terms of the quantity to scale and choice of whether or not to column normalize the penultimate layer activations in section C.2, where we find that the regularization strength is multi-faceted.

That is, we observe that the regularization strength equals the variance, when a practitioner chooses to scale the dropped-out activations in accord with standard dropout and column normalize the penultimate layer activations. Otherwise, the regularization strength equals the square of the coefficient of variation. Interestingly, if we compare and contrast the regularization strengths in sections 2.2, B.2, and C.2, then we find that the multi-faceted character of the regularization strength requires the model to be data-dependent, i.e., the non data-dependent model in matrix factorization is a matrix product of parameters.

---

[7]The point about the one-time cost of re-scale follows from the reduction of the predictions to canonical form, where we scale the fitted original parameters and turn them into the fitted re-scaled parameters before we generate predictions at test time.

Additionally, we explore the average behavior of generalized linear models with dropout in section D, where we generalize the decomposition of the expected loss under dropout, regularization term under dropout, and quadratic approximation of the regularization term under dropout in the seminal work of Wager et al. (2013). To do so, we handle the multiplicative nature of dropout in our derivations and explanations, whereas the work of Wager et al. (2013) adopted the works of Webb (1994) and Bishop (1995) in order to work on the zero mean additive case of dropout.

The upshot of this is that we are able to reason in sections D.2.1 and D.2.2 about the approximate average behavior in a way that resembles the exact average behavior in sections 2.1.1, B.1.1, and C.1.1. In particular, we find that the quadratic approximation of the expected loss under dropout is many-to-one in terms of the original dropped-out quantity, which implies that the dropout-induced regularization and optimization are ambiguous, when we think from the perspective of this approximate decomposition.

Correspondingly, we reformulate the approximate decomposition in terms of the quantity to scale and choice of whether or not to column normalize the weighted design matrix in section D.3, where we find that the regularization strength is multi-faceted in accord with sections 2.2 and C.2. Therefore, we have demonstrated that a single dropout distribution gives rise to two types of regularization strength in three contexts of machine learning, where the type of regularization strength depends on the choices that a practitioner makes in the learning process with dropout.

### 3.1 Open question on the equivalence of the output at test time

In contrast to section 2.3, we do not possess a closed-form solution in these contexts of machine learning. So, we leave open the question of whether or not the reformulations result in the same output at test time in sections B.4, C.4, and D.5.

Illustratively, we represent this with the transition from solid lines to dashed lines in Figures 3, 5, and 7, where we recall that Figure 1 uses solid lines throughout due to the results in section 2.3.1.

### 3.2 Penalty term under dropout

Nevertheless, we are able to compute the penalty term under dropout in sections B.5, C.5, and D.6, where we find that the penalty term under dropout depends on the data, parameters, and predictions at train time in a way that is similar to section 2.4. Then, we illustrate the penalty term under dropout in Figures 4, 6, and 8 such that they share the structure that we observe in Figure 2.

### 3.3 Towards a more general theory of the learning process with dropout

Our results are based on an examination of the first moment of the learning process with dropout in the contexts of linear regression, matrix factorization, fully-connected neural networks with dropout in the last layer, and generalized linear models. Accordingly, we propose to develop a more general theory in terms of the higher moments of the learning process with dropout in these contexts of machine learning and beyond.

Moreover, we did not intertwine our operational perspective of the nature of dropout with the gradient descent iterates. Thus, we suggest a juxtaposition of the higher moments and gradient descent iterates in accord with the works of Clara et al. (2024) and Zhang & Xu (2024).

### 3.4 Contextualization of prior work

Previously, we discussed part of the work of Srivastava et al. (2014) in the introduction and section 2.1, where they dropped out the design matrix based on $\mathcal{O}(mn)$ draws from a Bernoulli distribution in order to obtain a special case of Eq. 4. Then, they performed experiments based on a random sample from a Gaussian distribution $\mathcal{N}(1, \sigma^2)$, where the dropout distribution has mean $\mu = 1$ and tunable variance $\sigma^2$. Consequently, we find that these experiments were based on a degenerate regularization strength in which the square of the coefficient of variation reduces to the variance, i.e., $(\frac{\sigma}{\mu})^2 = \sigma^2$.

Furthermore, we recall that the Keras and TensorFlow machine learning libraries offer this version of Gaussian dropout, where the rate argument $p$ enables us to tune the variance $\sigma^2 = \frac{p}{1-p}$ (Chollet et al., 2015; Abadi et al., 2016). Thus, we conclude that the regularization strength is degenerate and the non-standard terms in the penalty term under dropout vanish, when we use a standard implementation of dropout in the Keras, TensorFlow, PyTorch, and MLX machine learning libraries, since these libraries offer either inverted dropout or Gaussian dropout (Chollet et al., 2015; Abadi et al., 2016; Paszke et al., 2019; Hannun et al., 2023).

Beyond the context of linear regression, the work of Cavazza et al. (2018) derived a special case of the decomposition in section B.1.1, where they dropped out either the left or right matrix factor based on $\mathcal{O}(r)$ draws from a Bernoulli distribution. Next, the work of Mianjy et al. (2018) derived a special case of the decomposition in section C.1.1, where they dropped out either the first or second layer weights of a two-layer fully connected neural network with no activation function based on $\mathcal{O}(n^{[1]})$ draws from a Bernoulli distribution. [8] Then, the work of Arora et al. (2021) continued this line of research, where they dropped out either the first layer activations or second layer weights of a two-layer fully connected neural network with a ReLU activation function based on $\mathcal{O}(n^{[1]})$ draws in the special case of inverted dropout.

In the context of generalized linear models, the work of Wager et al. (2013) leveraged the works of Webb (1994) and Bishop (1995) in order to derive a special case of the decomposition and regularization term under dropout in section D.2 as well as the quadratic approximation of the regularization term under dropout in section D.2.1. To do so, they dropped out the examples based on $\mathcal{O}(mn)$ draws in the special case of inverted dropout.

Next, the works of Helmbold & Long (2015; 2018) built off of these results, which led to the belief that the penalty term under dropout depends on neither the entire data nor the predictions at train time in the contexts of linear regression, generalized linear models, and matrix factorization. Then, the works of Wei et al. (2020), Blanchet et al. (2023), and Zhang & Xu (2024) extended this line of research into the contexts of fully-connected neural networks with dropout in multiple layers, distributionally robust optimization, and the modified gradient flow under dropout, respectively. Thus, we observe that there exist more generalizations to derive and explain based on the methods and insight that have been developed in this work.

## 3.5 Component-wise approach to dropout

Methodologically, we employed the component-wise approach to dropout in our derivations and explanations. The reason is that the standard and component-wise approaches result in the same decomposition. However, the component-wise approach is based on a diagonal random matrix and the standard approach is based on a dense rectangular random matrix, which implies that the former is more sample and memory efficient than the latter. [9]

Accordingly, we propose to conduct experiments, which explore the trade-off between a baseline of no dropout, a granular approach to dropout, and the standard approach to dropout in terms of the computational overhead, performance, and effectiveness of the regularization. For example, we could consider an experiment with a transformer-based model that takes an input in the shape of batch size, number of tokens, and number of components in the token space. Then, the component-wise approach would disregard the number of tokens and possibly the batch size as well, whereas the example-wise approach would disregard the number of components in the token space and possibly the batch size as well.

Similarly, we could consider other models, such as a diffusion-based model, where the granular approach would be with respect to the shape of the input to the model. In conclusion, the point of these experiments would be to determine how well granular versions of dropout perform in the frontier setting of machine learning and artificial intelligence, where our derivations and explanations trail the various empirical endeavors.

---

[8]The works of Cavazza et al. (2018) and Mianjy et al. (2018) re-scale the dropped-out model in the expected loss under dropout such that the results resemble the special case of inverted dropout.

[9]Notably, the research of Cavazza et al. (2018), Mianjy et al. (2018), and Arora et al. (2021) worked in terms of the component-wise approach to dropout, whereas the research of Srivastava et al. (2014) and Wager et al. (2013) worked in terms of the standard approach to dropout.

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

# A  Linear regression

In what follows, we derive the decomposition of the expected loss under dropout in sections A.1 and A.1.1. Next, we compute the gradient and Hessian of the expected loss under dropout in sections A.1.2 and A.1.3, where the former is with respect to the parameters and the latter is with respect to the re-scaled parameters. Then, we derive the equivalence of predictions at test time in section A.1.4, where we show that the invariant in these predictions is the square of the coefficient of variation $(\frac{\sigma}{\mu})^2$ of the dropout distribution.

## A.1  Derivation of the decomposition

To derive Eq. 4, we begin with Eq. 3

$$\mathbb{E}[J(\theta)] = \mathbb{E}[||Y - X\delta\theta||_2^2] = \sum_{i=1}^{m} \mathbb{E}[(Y - X\delta\theta)_i^2],$$

which expands to

$$\sum_{i=1}^{m} \operatorname{Var}(Y_i - \sum_{j=1}^{n} X_{ij}d_j\theta_j) + \mathbb{E}[Y_i - \sum_{k=1}^{n} X_{ik}d_k\theta_k]^2$$

due to properties of the variance. Next, we have that this expansion simplifies to

$$\sum_{i=1}^{m}(Y - \mu X\theta)_i^2 + \sigma^2 \sum_{j=1}^{n} \theta_j^2 \sum_{i=1}^{m} X_{ij}^2,$$

which is equal to

$$||Y - \mu X\theta||_2^2 + \sum_{j=1}^{n}(\sigma||X_j||_2\theta_j)^2.$$

Thus, we conclude that Eq. 3 decomposes into Eq. 4, since

$$||\Gamma\theta||_2^2 = ||\sigma G\theta||_2^2 = \sum_{j=1}^{n}(\sigma||X_j||_2\theta_j)^2,$$

where the inverse of the column normalization matrix $G = \operatorname{diag}(||X_1||_2, \ldots, ||X_n||_2)$. Moreover, we observe that the expected loss under dropout decomposes into a ridge regression problem

$$\mathbb{E}[J(\theta)] = ||Y - \mu X\theta||_2^2 + \sigma^2||\theta||_2^2,$$

when we normalize the columns of the design matrix such that the Tikhonov matrix $\Gamma = \sigma G = \sigma \mathbb{I}$, i.e., $||X_j||_2 = 1$ for $j = 1, \ldots, n$.

### A.1.1  Derivation of the decomposition with column normalization

Throughout this work, we tend to overload the notation $X$, when we column normalize the design matrix. The reason is that the more pedantic derivation and explanation turns out to be similar.

Namely, if we consider the expected loss under dropout

$$\mathbb{E}[J(\theta)] = \mathbb{E}[||Y - A\delta\theta||_2^2],$$

where $A = XG^{-1}$ denotes the design matrix with column normalization such that

$$G^{-1} = \text{diag}\left(\frac{1}{||X_1||_2}, \ldots, \frac{1}{||X_n||_2}\right)$$

is the column normalization matrix and $||A_j||_2 = \frac{||X_j||_2}{||X_j||_2} = 1$ for $j = 1, \ldots, n$. Then, we have that the expected loss under dropout decomposes into a ridge regression problem

$$\mathbb{E}[J(\theta)] = ||Y - \mu A\theta||_2^2 + \sigma^2 ||\theta||_2^2,$$

where the proof follows from the proof in section A.1 such that

$$\sum_{j=1}^{n}(\sigma ||A_j||_2 \theta_j)^2 = \sigma^2 ||\theta||_2^2.$$

Consequently, we interchange the notation from $X$ to $A$ and vice versa, when we need to be more or less pedantic in the derivations and explanations in this work.

### A.1.2 Derivatives of the expected loss under dropout with respect to the parameters

From Eq. 4, we have that

$$\frac{1}{2}\nabla_\theta(\mathbb{E}[J(\theta)]) = \frac{1}{2}\nabla_\theta(||Y - \mu X\theta||_2^2) + \frac{1}{2}\nabla_\theta(||\Gamma\theta||_2^2),$$

where the decomposition enables us to compute the gradient of the expected loss under dropout with respect to the parameters. In particular, we have that

$$\frac{1}{2}\nabla_\theta(\mathbb{E}[J(\theta)]) = \mu X^t(\mu X\theta - Y) + \Gamma^t\Gamma\theta = \mu X^t(\mu X\theta - Y) + \sigma^2 G^t G\theta,$$

where the square of the mean in the term $\mu^2 X^t X\theta$ implies that we must reformulate the gradient of the expected loss under dropout with respect to the parameters in terms of the re-scaled design matrix.

Consequently, if we reformulate, then we have that

$$\frac{1}{2}\nabla_\theta(\mathbb{E}[J(\theta)]) = \tilde{X}^t(\tilde{X}\theta - Y) + \left(\frac{\sigma}{\mu}\right)^2 \tilde{G}^t \tilde{G}\theta,$$

which implies the following normal equations

$$\left(\tilde{X}^t\tilde{X} + \left(\frac{\sigma}{\mu}\right)^2 \tilde{G}^t\tilde{G}\right)\theta = \tilde{X}^t Y.$$

Moreover, if we column normalize and reformulate, then we have that

$$\frac{1}{2}\nabla_\theta(\mathbb{E}[J(\theta)]) = \tilde{X}^t(\tilde{X}\theta - Y) + \sigma^2\theta,$$

which implies the following normal equations

$$(\tilde{X}^t\tilde{X} + \sigma^2\mathbb{I})\theta = \tilde{X}^t Y.$$

Correspondingly, if we reformulate, then we have that the Hessian of the expected loss under dropout with respect to the parameters is

$$\frac{1}{2}D(\nabla_\theta(\mathbb{E}[J(\theta)])) = \tilde{X}^t\tilde{X} + \left(\frac{\sigma}{\mu}\right)^2 \tilde{G}^t\tilde{G}.$$

Otherwise, if we column normalize and reformulate, then we have that the Hessian of the expected loss under dropout with respect to the parameters is

$$\frac{1}{2}D(\nabla_\theta(\mathbb{E}[J(\theta)])) = \tilde{X}^t\tilde{X} + \sigma^2\mathbb{I}.$$

Therefore, we have that the dropout-induced geometry retains the connections that we found in section 2.2.1.

### A.1.3 Derivatives of the expected loss under dropout with respect to the re-scaled parameters

Next, we want to show that the work in section A.1.2 persists, when we compute the derivatives of the expected loss under dropout with respect to the re-scaled parameters. From Eq. 4, we have that

$$\frac{1}{2}\nabla_{\tilde{\theta}}(\mathbb{E}[J(\theta)]) = \frac{1}{2}\nabla_{\tilde{\theta}}(||Y - \mu X\theta||_2^2) + \frac{1}{2}\nabla_{\tilde{\theta}}(||\Gamma\theta||_2^2),$$

where the decomposition enables us to compute the gradient of the expected loss under dropout with respect to the re-scaled parameters, when we reformulate the decomposition in terms of the re-scaled parameters.

Namely, we have that

$$\frac{1}{2}\nabla_{\tilde{\theta}}(\mathbb{E}[J(\theta)]) = X^t(X\tilde{\theta} - Y) + \frac{1}{\mu^2}\Gamma^t\Gamma\tilde{\theta} = X^t(X\tilde{\theta} - Y) + (\frac{\sigma}{\mu})^2 G^t G\tilde{\theta},$$

which implies the following normal equations

$$(X^t X + (\frac{\sigma}{\mu})^2 G^t G)\tilde{\theta} = X^t Y.$$

Moreover, if we column normalize and reformulate, then we have that

$$\frac{1}{2}\nabla_{\tilde{\theta}}(\mathbb{E}[J(\theta)]) = X^t(X\tilde{\theta} - Y) + (\frac{\sigma}{\mu})^2\tilde{\theta},$$

which implies the following normal equations

$$(X^t X + (\frac{\sigma}{\mu})^2\mathbb{I})\tilde{\theta} = X^t Y.$$

Correspondingly, if we reformulate, then we have that the Hessian of the expected loss under dropout with respect to the re-scaled parameters is

$$\frac{1}{2}D(\nabla_{\tilde{\theta}}(\mathbb{E}[J(\theta)])) = X^t X + (\frac{\sigma}{\mu})^2 G^t G,$$

Otherwise, if we column normalize and reformulate, then we have that the Hessian of the expected loss under dropout with respect to the re-scaled parameters is

$$\frac{1}{2}D(\nabla_{\tilde{\theta}}(\mathbb{E}[J(\theta)])) = X^t X + (\frac{\sigma}{\mu})^2\mathbb{I}.$$

Therefore, we conclude that the work in section A.1.2 persists, when we compute the derivatives of the expected loss under dropout with respect to the re-scaled parameters. Moreover, we have that the dropout-induced geometry retains the connections that we found in section 2.2.2.

### A.1.4 Derivation of the equivalence of predictions at test time

To demonstrate the equivalence of Eq. 6, Eq. 8, Eq. 10, and Eq. 12, we recall that Eq. 6 and Eq. 10 correspond to a prediction from a particular form of linear regression with Tikhonov regularization

$$x \cdot (X^t X + (\frac{\sigma}{\mu})^2 G^t G)^{-1} X^t Y,$$

where $x$ denotes a test example without column normalization, $X$ denotes the design matrix without column normalization, and $G^t G = \text{diag}(||X_1||_2^2, \ldots, ||X_n||_2^2) = G^2$ denotes the Gram matrix of the inverse of the column normalization matrix. Then, we recall that Eq. 8 and Eq. 12 correspond to a prediction from a particular form of ridge regression

$$a \cdot (A^t A + (\frac{\sigma}{\mu})^2\mathbb{I})^{-1} A^t Y,$$

where $a = G^{-1}x$ denotes a test example with column normalization and $A = XG^{-1}$ denotes the design matrix with column normalization; see the discussion on the interchange of notation in section A.1.1.

Subsequently, we claim that Eq. 6 and Eq. 10 are equivalent to Eq. 8 and Eq. 12, since a prediction from the particular form of linear regression with Tikhonov regularization is equivalent to a prediction from the particular form of ridge regression

$$x \cdot (X^t X + (\frac{\sigma}{\mu})^2 G^t G)^{-1} X^t Y = a \cdot (A^t A + (\frac{\sigma}{\mu})^2 \mathbb{I})^{-1} A^t Y.$$

To prove this, we begin with a prediction from the particular form of ridge regression

$$a \cdot (A^t A + (\frac{\sigma}{\mu})^2 \mathbb{I})^{-1} A^t Y = G^{-1}x \cdot (G^{-1}X^t X G^{-1} + (\frac{\sigma}{\mu})^2 \mathbb{I})^{-1} G^{-1} X^t Y.$$

Then, we have that this prediction is equivalent to

$$G^{-1}x \cdot G(X^t X + (\frac{\sigma}{\mu})^2 G^2)^{-1} G G^{-1} X^t Y = x \cdot (X^t X + (\frac{\sigma}{\mu})^2 G^t G)^{-1} X^t Y.$$

Thus, we prove the claim, which implies that the predictions in Eq. 6, Eq. 8, Eq. 10, and Eq. 12 are equivalent. Moreover, we observe that the invariant in these predictions is the square of the coefficient of variation $(\frac{\sigma}{\mu})^2$ of the dropout distribution, which appears as the scalar multiple of the Gram matrix of the inverse of the column normalization matrix and the ridge parameter.

## B    Matrix factorization

Next, we examine the problem of matrix factorization. Here,

$$J(A, B) = \left| \left| X - AB \right| \right|_F^2 \tag{14}$$

denotes the squared error loss function, where the norm is the Frobenius norm. Specifically, the term $X$ is a target matrix with shape $m \times n$. Then, the term $A$ is the left matrix factor with shape $m \times r$ and the term $B$ is the right matrix factor with shape $r \times n$, where $r$ is a hyper-parameter that balances reconstruction accuracy and model complexity. Overall, the objective is to find the left and right matrix factors, which minimize the squared error loss function. [10]

### B.1    Matrix factorization with dropout

To study matrix factorization with dropout, we draw independent and identically distributed random variables $d_j \overset{\text{iid}}{\sim} \mathfrak{D}$ from a probability distribution that has finite but non-zero mean $\mathbb{E}[d_j] = \mu$ and finite variance $\text{Var}(d_j) = \sigma^2$, where $j = 1, \ldots, r$. Next, we construct a dropout mask $\delta = \text{diag}(d_1, \ldots, d_r)$, which is a diagonal matrix with the dropout random variables on the diagonal. Then, we form the composite dropped-out product

$$A\delta B = (A\delta)B = A(\delta B),$$

where $A\delta$ and $\delta B$ correspond to the dropped-out left and right matrix factors, respectively.

#### B.1.1    Expected loss under dropout

Accordingly, we form the squared error loss under dropout

$$J(A, B) = \left| \left| X - A\delta B \right| \right|_F^2.$$

Next, we consider the expected loss under dropout

$$\mathbb{E}[J(A, B)] = \mathbb{E}[\left| \left| X - A\delta B \right| \right|_F^2], \tag{15}$$

---

[10]In this unsupervised learning context, we treat the target matrix as a collection of labels and the product of the left and right matrix factors as a non-data-dependent model of the collection of labels, which we call a supervised learning relaxation.

since this expectation value enables us to analyze the average behavior of matrix factorization with dropout. Then, we have that Eq. 15 decomposes into a sum of a squared error loss term and a non-data-dependent regularization term

$$\mathbb{E}[J(A,B)] = \left|\left|X - \mu AB\right|\right|_F^2 + \sigma^2 \sum_{k=1}^{r} \left|\left|A_k\right|\right|_2^2 \left|\left|B_k^t\right|\right|_2^2, \tag{16}$$

where the regularization term depends on the columns of the left matrix factor and rows of the right matrix factor; see the details in section B.3.

Thereupon, we observe that Eq. 16 is many-to-one in terms of the original dropped-out quantity. The implication of this property of the average behavior is that our interpretation of the dropout-induced regularization and optimization is ambiguous, when we think from the perspective of Eq. 16.

## B.2   Reformulation of the decomposition

Correspondingly, we reformulate Eq. 16 in terms of the matrix factor to scale, which is consistent with our approach in section 2.2. To this end, we begin with a practitioner, who chooses to scale the left matrix factor. Next, we map the original left matrix factor to the re-scaled left matrix factor $\tilde{A} = \mu A = \mathbb{E}[A\delta]$. Then, we have that

$$J(\tilde{A}, B) = \left|\left|X - \tilde{A}B\right|\right|_F^2 + (\frac{\sigma}{\mu})^2 \sum_{k=1}^{r} \left|\left|\tilde{A}_k\right|\right|_2^2 \left|\left|B_k^t\right|\right|_2^2, \tag{17}$$

which is a squared error loss function of the re-scaled left matrix factor and original right matrix factor with a non-data-dependent regularization term.

Ensuingly, we consider a practitioner, who chooses to scale the right matrix factor. Next, we map the original right matrix factor to the re-scaled right matrix factor $\tilde{B} = \mu B = \mathbb{E}[\delta B]$. Then, we have that

$$J(A, \tilde{B}) = \left|\left|X - A\tilde{B}\right|\right|_F^2 + (\frac{\sigma}{\mu})^2 \sum_{k=1}^{r} \left|\left|A_k\right|\right|_2^2 \left|\left|\tilde{B}_k^t\right|\right|_2^2, \tag{18}$$

which is a squared error loss function of the original left matrix factor and re-scaled right matrix factor with a non-data-dependent regularization term.

Accordingly, we find that the regularization strength equals the square of the coefficient of variation $\lambda = (\frac{\sigma}{\mu})^2$, when we think about the nature of dropout from the perspective of either Eq. 17 or Eq. 18. Moreover, we observe that the optimization problem of interest is in terms of an original matrix factor and a re-scaled matrix factor and there is symmetry in the appearance of the original matrix factor and the re-scaled matrix factor in the loss term and regularization term.

Furthermore, we reformulate the gradient and Hessian of the expected loss under dropout with respect to the left matrix factor, re-scaled left matrix factor, right matrix factor, and re-scaled right matrix factor in sections B.3.1, B.3.2, B.3.3, and B.3.4, respectively. Ultimately, we find that the connections in the dropout-induced regularization persist in the geometry.

## B.3   Derivation of the decomposition

To derive Eq. 16, we begin with Eq. 15

$$\mathbb{E}[J(A,B)] = \mathbb{E}[\left|\left|X - A\delta B\right|\right|_F^2] = \sum_{i=1}^{m} \sum_{j=1}^{n} \mathbb{E}[(X - A\delta B)_{ij}^2],$$

which expands to

$$\sum_{i=1}^{m} \sum_{j=1}^{n} \text{Var}(X_{ij} - \sum_{k=1}^{r} A_{ik} d_k B_{kj}) + \mathbb{E}[X_{ij} - \sum_{l=1}^{r} A_{il} d_l B_{lj}]^2$$

due to properties of the variance. Next, we have that this expansion simplifies to

$$\sum_{i=1}^{m}\sum_{j=1}^{n}(X-\mu AB)_{ij}^2 + \sigma^2\sum_{k=1}^{r}\sum_{i=1}^{m}A_{ik}^2\sum_{j=1}^{n}B_{kj}^2,$$

which is equal to

$$||X-\mu AB||_F^2 + \sigma^2\sum_{k=1}^{r}||A_k||_2^2||B_k^t||_2^2.$$

Thus, we conclude that Eq. 15 decomposes into Eq. 16.

### B.3.1 Derivatives of the expected loss under dropout with respect to the left matrix factor

From Eq. 16, we have that

$$\frac{1}{2}\nabla_A(\mathbb{E}[J(A,B)]) = \frac{1}{2}\nabla_A(||X-\mu AB||_F^2) + \frac{1}{2}\nabla_A(\sigma^2\sum_{k=1}^{r}||A_k||_2^2||B_k^t||_2^2),$$

where the decomposition enables us to compute the gradient of the expected loss under dropout with respect to the left matrix factor. In particular, we have that

$$\frac{1}{2}\nabla_A(\mathbb{E}[J(A,B)]) = \mu(\mu AB - X)B^t + \sigma^2 A\operatorname{diag}\left(||B_1^t||_2,\ldots,||B_r^t||_2\right)^2,$$

where the square of the mean in the term $\mu^2 ABB^t$ implies that we can only reformulate the gradient of the expected loss under dropout with respect to the left matrix factor in terms of the re-scaled right matrix factor.

Consequently, if we reformulate, then we have that

$$\frac{1}{2}\nabla_A(\mathbb{E}[J(A,B)]) = (A\tilde{B} - X)\tilde{B}^t + (\frac{\sigma}{\mu})^2 A\operatorname{diag}\left(||\tilde{B}_1^t||_2,\ldots,||\tilde{B}_r^t||_2\right)^2.$$

Correspondingly, if we reformulate, then we have that the Hessian of the expected loss under dropout with respect to the left matrix factor is

$$\frac{1}{2}D(\nabla_A(\mathbb{E}[J(A,B)])) = \mathbb{I}\otimes\left(\tilde{B}\tilde{B}^t + (\frac{\sigma}{\mu})^2\operatorname{diag}\left(||\tilde{B}_1^t||_2,\ldots,||\tilde{B}_r^t||_2\right)^2\right).$$

Therefore, we have that the dropout-induced geometry retains the connections that we found in section B.2.

### B.3.2 Derivatives of the expected loss under dropout with respect to the re-scaled left matrix factor

Next, we want to show that the work in section B.3.1 persists, when we compute the derivatives of the expected loss under dropout with respect to the re-scaled left matrix factor. From Eq. 16, we have that

$$\frac{1}{2}\nabla_{\tilde{A}}(\mathbb{E}[J(A,B)]) = \frac{1}{2}\nabla_{\tilde{A}}(||X-\mu AB||_F^2) + \frac{1}{2}\nabla_{\tilde{A}}(\sigma^2\sum_{k=1}^{r}||A_k||_2^2||B_k^t||_2^2),$$

where the decomposition enables us to compute the gradient of the expected loss under dropout with respect to the re-scaled left matrix factor, when we reformulate the decomposition in terms of the re-scaled left matrix factor. In particular, we have that

$$\frac{1}{2}\nabla_{\tilde{A}}(\mathbb{E}[J(A,B)]) = (\tilde{A}B - X)B^t + (\frac{\sigma}{\mu})^2\tilde{A}\operatorname{diag}\left(||B_1^t||_2,\ldots,||B_r^t||_2\right)^2.$$

Correspondingly, if we reformulate, then we have that the Hessian of the expected loss under dropout with respect to the re-scaled left matrix factor is

$$\frac{1}{2}D(\nabla_{\tilde{A}}(\mathbb{E}[J(A,B)])) = \mathbb{I}\otimes\left(BB^t + (\frac{\sigma}{\mu})^2\operatorname{diag}\left(||B_1^t||_2,\ldots,||B_r^t||_2\right)^2\right).$$

Therefore, we conclude that the work in section B.3.1 persists, when we compute the derivatives of the expected loss under dropout with respect to the re-scaled left matrix factor. Moreover, we have that the dropout-induced geometry retains the connections that we found in section B.2.

### B.3.3 Derivatives of the expected loss under dropout with respect to the right matrix factor

From Eq. 16, we have that

$$\frac{1}{2}\nabla_B(\mathbb{E}[J(A,B)]) = \frac{1}{2}\nabla_B(||X - \mu AB||_F^2) + \frac{1}{2}\nabla_B(\sigma^2 \sum_{k=1}^{r} ||A_k||_2^2 ||B_k^t||_2^2),$$

where the decomposition enables us to compute the gradient of the expected loss under dropout with respect to the right matrix factor. In particular, we have that

$$\frac{1}{2}\nabla_B(\mathbb{E}[J(A,B)]) = \mu A^t(\mu AB - X) + \sigma^2 \operatorname{diag}\left(||A_1||_2, \ldots, ||A_r||_2\right)^2 B,$$

where the square of the mean in the term $\mu^2 A^t AB$ implies that we can only reformulate the gradient of the expected loss under dropout with respect to the right matrix factor in terms of the re-scaled left matrix factor.

Consequently, if we reformulate, then we have that

$$\frac{1}{2}\nabla_B(\mathbb{E}[J(A,B)]) = \tilde{A}^t(\tilde{A}B - X) + (\frac{\sigma}{\mu})^2 \operatorname{diag}\left(||\tilde{A}_1||_2, \ldots, ||\tilde{A}_r||_2\right)^2 B.$$

Correspondingly, if we reformulate, then we have that the Hessian of the expected loss under dropout with respect to the right matrix factor is

$$\frac{1}{2}D(\nabla_B(\mathbb{E}[J(A,B)])) = (\tilde{A}^t\tilde{A} + (\frac{\sigma}{\mu})^2 \operatorname{diag}\left(||\tilde{A}_1||_2, \ldots, ||\tilde{A}_r||_2\right)^2) \otimes \mathbb{I}.$$

Therefore, we have that the dropout-induced geometry retains the connections that we found in section B.2.

### B.3.4 Derivatives of the expected loss under dropout with respect to the re-scaled right matrix factor

Next, we want to show that the work in section B.3.3 persists, when we compute the derivatives of the expected loss under dropout with respect to the re-scaled right matrix factor. From Eq. 16, we have that

$$\frac{1}{2}\nabla_{\tilde{B}}(\mathbb{E}[J(A,B)]) = \frac{1}{2}\nabla_{\tilde{B}}(||X - \mu AB||_F^2) + \frac{1}{2}\nabla_{\tilde{B}}(\sigma^2 \sum_{k=1}^{r} ||A_k||_2^2 ||B_k^t||_2^2),$$

where the decomposition enables us to compute the gradient of the expected loss under dropout with respect to the re-scaled right matrix factor, when we reformulate the decomposition in terms of the re-scaled right matrix factor. In particular, we have that

$$\frac{1}{2}\nabla_{\tilde{B}}(\mathbb{E}[J(A,B)]) = A^t(A\tilde{B} - X) + (\frac{\sigma}{\mu})^2 \operatorname{diag}\left(||A_1^t||_2, \ldots, ||A_r^t||_2\right)^2 \tilde{B}.$$

Correspondingly, if we reformulate, then we have that the Hessian of the expected loss under dropout with respect to the re-scaled right matrix factor is

$$\frac{1}{2}D(\nabla_{\tilde{B}}(\mathbb{E}[J(A,B)])) = (A^tA + (\frac{\sigma}{\mu})^2 \operatorname{diag}\left(||A_1^t||_2, \ldots, ||A_r^t||_2\right)^2) \otimes \mathbb{I}.$$

Therefore, we conclude that the work in section B.3.3 persists, when we compute the derivatives of the expected loss under dropout with respect to the re-scaled right matrix factor. Moreover, we have that the dropout-induced geometry retains the connections that we found in section B.2.

### B.4 Dual interpretations

From section B.2, we found that there exist multiple interpretations of the average behavior of matrix factorization with dropout such that each interpretation details the regularization strength and optimization problem to solve based on the ways in which a practitioner is able to scale a matrix factor. However, we do not possess a closed form solution in this context of machine learning, so we leave open the question of whether or not the reformulations result in the same matrix factorization; see Figure 3.

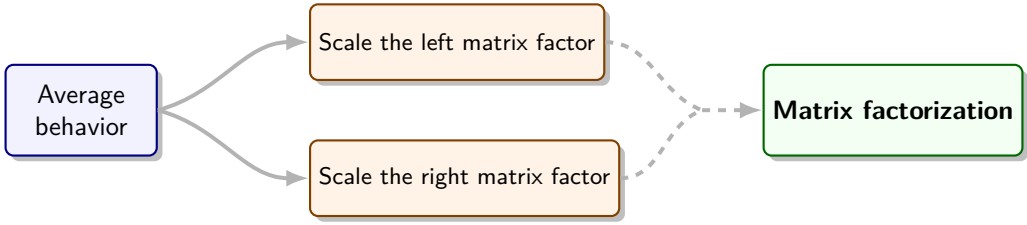

Figure 3: We reformulate the average behavior of matrix factorization with dropout in two different ways in order to disambiguate the dropout-induced regularization and optimization. Then, we leave open the question of whether or not the reformulations result in the same output matrix factorization.

### B.5 Penalty term under dropout

Nevertheless, we are able to explore the penalty term under dropout and analyze the ways in which it differs from the base penalty term, which provides insights into the train time behavior of matrix factorization with dropout. To begin, we define the base penalty term to be

$$\sigma^2 \sum_{k=1}^{r} \left|\left|A_k\right|\right|_2^2 \left|\left|B_k^t\right|\right|_2^2,$$

which depends on the parameters in the matrix factors and not the data in the target matrix. Then, we compute the penalty term under dropout

$$\mathbb{E}[J(A,B)] - J(A,B) = 2(1-\mu)\langle X, AB\rangle_F + (\mu^2 - 1)\left|\left|AB\right|\right|_F^2 + \sigma^2 \sum_{k=1}^{r} \left|\left|A_k\right|\right|_2^2 \left|\left|B_k^t\right|\right|_2^2, \tag{19}$$

which is the difference between the decomposition in Eq. 16 and the squared error loss function in Eq. 14.

From the right-hand side of Eq. 19, we have that the penalty term under dropout includes two regularization terms that go beyond the base regularization term.

- In particular, the first term corresponds to a scalar multiple of the Frobenius inner product between the target matrix and matrix factorization at train time $2(1-\mu)\langle X, AB\rangle_F$, which shows that the penalty term under dropout depends on the overlap between the target matrix and matrix factorization at train time.

- Then, the second term corresponds to a scalar multiple of the square of the Frobenius norm of the matrix factorization at train time $(\mu^2 - 1)\left|\left|AB\right|\right|_F^2$, which shows that the penalty term under dropout depends on the size of the matrix factorization at train time.

Interestingly, we observe that the two additional terms on the right-hand side of Eq. 19 vanish, when a practitioner picks a dropout distribution such that $\mu = 1$, i.e., inverted dropout or Gaussian dropout. Otherwise, the entire penalty term under dropout remains intact, when a practitioner picks a dropout distribution such that $\mu \neq 1$, i.e., a Bernoulli distribution with $\mu = p \in (0, 1)$; see Figure 4.

## C Fully-connected neural network

Then, we examine the problem of regression with a fully-connected neural network without any bias terms. Here,

$$J(W^{[1]}, \cdots, W^{[r]}) = \left|\left|Y - a^{[r]}\right|\right|_2^2 = \left|\left|Y - z^{[r]}\right|\right|_2^2 = \left|\left|Y - a^{[r-1]}W^{[r]}\right|\right|_2^2 \tag{20}$$

denotes the squared error loss function. Specifically, the term $Y$ is the collection of $m$ targets. Next, the term $a^{[r]}$ is the final layer activations with $m$ predictions, and the term $z^{[r]}$ is the logits with the same $m$

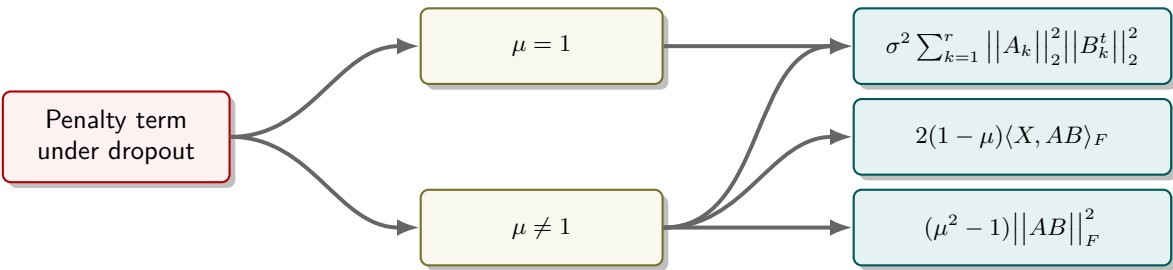

Figure 4: In the case of $\mu = 1$, the penalty term under dropout corresponds to the base penalty term. Otherwise, the penalty term under dropout generalizes the base penalty term with the addition of two terms, which depend on the overlap between the target matrix and matrix factorization at train time as well as the size of the matrix factorization at train time.

predictions. Then, the term $a^{[r-1]}$ is the penultimate layer activations with $m$ examples and $n^{[r-1]}$ derived features, and the term $W^{[r]}$ is the final layer weights with $n^{[r-1]}$ parameters.

Recursively, $z^{[l]} = a^{[l-1]}W^{[l]}$ denotes the $l$th layer pre-activations with $m$ examples and $n^{[l]}$ derived features, which is the matrix product of the previous layer activations $a^{[l-1]}$ with $m$ examples and $n^{[l-1]}$ derived features and current layer weights $W^{[l]}$ with $n^{[l-1]}n^{[l]}$ parameters. Then, $a^{[l]} = f(z^{[l]})$ denotes the $l$th layer activations, which is the image of the element-wise activation function $f$ of the $l$th layer pre-activations $z^{[l]}$, where $l = 1, \ldots, r - 1$ such that $a^{[0]} = X$ is the design matrix with $m$ examples and $n$ original features. Overall, the objective is to find the collection of weights, which minimize the squared error loss function.

### C.1 Fully-connected neural network with dropout in the last layer

To study a fully-connected neural network with dropout in the last layer, we draw independent and identically distributed random variables $d_j \overset{\text{iid}}{\sim} \mathfrak{D}$ from a probability distribution that has finite but non-zero mean $\mathbb{E}[d_j] = \mu$ and finite variance $\text{Var}(d_j) = \sigma^2$, where $j = 1, \ldots, n^{[r-1]}$. Next, we construct a dropout mask $\delta = \text{diag}(d_1, \ldots, d_{n^{[r-1]}})$, which is a diagonal matrix with the dropout random variables on the diagonal. Then, we form the composite dropped-out product

$$a^{[r-1]}\delta W^{[r]} = (a^{[r-1]}\delta)W^{[r]} = a^{[r-1]}(\delta W^{[r]}),$$

where $a^{[r-1]}\delta$ and $\delta W^{[r]}$ correspond to the dropped-out penultimate layer activations and last layer weights, respectively.

#### C.1.1 Expected loss under dropout

Accordingly, we form the squared error loss under dropout

$$J(W^{[1]}, \cdots, W^{[r]}) = \left|\left|Y - a^{[r-1]}\delta W^{[r]}\right|\right|_2^2.$$

Next, we consider the expected loss under dropout

$$\mathbb{E}[J(W^{[1]}, \cdots, W^{[r]})] = \mathbb{E}[\left|\left|Y - a^{[r-1]}\delta W^{[r]}\right|\right|_2^2], \tag{21}$$

since this expectation value enables us to analyze the average behavior of a fully-connected neural network with dropout in the last layer. Then, we have that Eq. 21 decomposes into a sum of a squared error loss term and a data-dependent Tikhonov regularization term

$$\mathbb{E}[J(W^{[1]}, \cdots, W^{[r]})] = \left|\left|Y - \mu a^{[r-1]}W^{[r]}\right|\right|_2^2 + \left|\left|\Gamma W^{[r]}\right|\right|_2^2, \tag{22}$$

where the Tikhonov matrix is a scalar multiple $\sigma$ of a diagonal matrix G such that

$$(G)_{jj} = \text{diag}(||a_1^{[r-1]}||_2, \ldots, ||a_{n^{[r-1]}}^{[r-1]}||_2)_{jj} = \left|\left|a_j^{[r-1]}\right|\right|_2,$$

$a_j^{[r-1]}$ denotes the $j$th column of the penultimate layer activations, and $j = 1, \ldots, n^{[r-1]}$; see the details in section C.3.

Thereupon, we observe that Eq. 22 is many-to-one in terms of the original dropped-out quantity. The implication of this property of the average behavior is that our interpretation of the dropout-induced regularization and optimization is ambiguous, when we think from the perspective of Eq. 22.

## C.2 Reformulation of the decomposition

Correspondingly, we reformulate Eq. 22 in terms of the quantity to scale and choice of whether or not to column normalize the penultimate layer activations, which is consistent with our approach in section 2.2. That is, we consider a practitioner, who chooses to scale the dropped-out activations in accord with standard dropout in section C.2.1 and the dropped-out weights in section C.2.2.

### C.2.1 Scale the activations

To carry this out, we begin with a practitioner, who chooses to scale the dropped-out activations in accord with standard dropout. Next, we map the original penultimate layer activations to the re-scaled penultimate layer activations $\tilde{a}^{[r-1]} = \mu a^{[r-1]} = \mathbb{E}[a^{[r-1]}\delta]$. Then, we have that the decomposition becomes

$$\mathbb{E}[J(W^{[1]}, \cdots, W^{[r]})] = ||Y - \tilde{a}^{[r-1]}W^{[r]}||_2^2 + (\frac{\sigma}{\mu})^2||\tilde{G}W^{[r]}||_2^2, \tag{23}$$

where $(\tilde{G})_{jj} = ||\mu a_j^{[r-1]}||_2 = ||\tilde{a}_j^{[r-1]}||_2$ for $j = 1, \ldots, n^{[r-1]}$.

Accordingly, we find that the regularization strength equals the square of the coefficient of variation $\lambda = (\frac{\sigma}{\mu})^2$, when we think about the nature of dropout from the perspective of Eq. 23. Moreover, we observe that the optimization problem of interest is in terms of the original weights and there is symmetry in the appearance of the re-scaled penultimate layer activations in the loss term and regularization term.

Similarly, we consider a practitioner, who chooses to scale the dropped-out activations and column normalize the penultimate layer activations such that $||a_j^{[r-1]}||_2 = 1$ for $j = 1, \ldots, n^{[r-1]}$. Then, we have that the decomposition becomes

$$\mathbb{E}[J(W^{[1]}, \cdots, W^{[r]})] = ||Y - \tilde{a}^{[r-1]}W^{[r]}||_2^2 + \sigma^2||W^{[r]}||_2^2. \tag{24}$$

Consequently, we find that the regularization strength equals the variance $\lambda = \sigma^2$, when we think about the nature of dropout from the perspective of Eq. 24. Moreover, we observe that the optimization problem of interest is in terms of the original weights and there is asymmetry in the appearance of the re-scaled penultimate layer activations in the loss term and regularization term.

Additionally, we reformulate the gradient and Hessian of the expected loss under dropout with respect to the last layer weights in section C.3.1. These computations show that the connections in the dropout-induced regularization persist in the geometry.

### C.2.2 Scale the weights

Ensuingly, we consider a practitioner, who chooses to scale the dropped-out weights. Next, we map the original last layer weights to the re-scaled last layer weights $\tilde{W}^{[r]} = \mu W^{[r]} = \mathbb{E}[\delta W^{[r]}]$. Then, we have that the decomposition becomes

$$J(W^{[1]}, \cdots, \tilde{W}^{[r]}) = ||Y - a^{[r-1]}\tilde{W}^{[r]}||_2^2 + (\frac{\sigma}{\mu})^2||G\tilde{W}^{[r]}||_2^2, \tag{25}$$

which is a squared error loss function of the original weights and re-scaled last layer weights with a Tikhonov regularization term.

Accordingly, we find that the regularization strength equals the square of the coefficient of variation $\lambda = (\frac{\sigma}{\mu})^2$, when we think about the nature of dropout from the perspective of Eq. 25. Moreover, we observe that the

optimization problem of interest is in terms of the original weights and re-scaled last layer weights and there is symmetry in the appearance of the re-scaled last layer weights in the loss term and regularization term.

Similarly, we consider a practitioner, who chooses to scale the dropped-out weights and column normalize the penultimate layer activations. Then, we have that the decomposition becomes

$$J(W^{[1]}, \cdots, \tilde{W}^{[r]}) = ||Y - a^{[r-1]}\tilde{W}^{[r]}||_2^2 + (\frac{\sigma}{\mu})^2 ||\tilde{W}^{[r]}||_2^2, \tag{26}$$

which is a ridge regression problem in terms of the original weights and re-scaled last layer weights.

Consequently, we find that the regularization strength equals the square of the coefficient of variation $\lambda = (\frac{\sigma}{\mu})^2$, when we think about the nature of dropout from the perspective of Eq. 26. Moreover, we observe that the optimization problem of interest is in terms of the original weights and re-scaled last layer weights and there is symmetry in the appearance of the re-scaled last layer weights in the loss term and regularization term.

Furthermore, we reformulate the gradient and Hessian of the expected loss under dropout with respect to the re-scaled last layer weights in section C.3.2. These computations show that the connections in the dropout-induced regularization persist in the geometry.

## C.3 Derivation of the decomposition

To derive Eq. 22, we begin with Eq. 21

$$\mathbb{E}[J(W^{[1]}, \cdots, W^{[r]})] = \mathbb{E}[||Y - a^{[r-1]}\delta W^{[r]}||_2^2] = \sum_{i=1}^{m} \mathbb{E}[(Y - a^{[r-1]}\delta W^{[r]})_i^2],$$

which expands to

$$\sum_{i=1}^{m} \text{Var}(Y_i - \sum_{j=1}^{n^{[r-1]}} a_{ij}^{[r-1]} d_j W_j^{[r]}) + \mathbb{E}[Y_i - \sum_{k=1}^{n^{[r-1]}} a_{ik}^{[r-1]} d_k W_k^{[r]}]^2$$

due to properties of the variance. Next, we have that this expansion simplifies to

$$\sum_{i=1}^{m} (Y - \mu a^{[r-1]} W^{[r]})_i^2 + \sigma^2 \sum_{j=1}^{n^{[r-1]}} (W_j^{[r]})^2 \sum_{i=1}^{m} (a_{ij}^{[r-1]})^2,$$

which is equal to

$$||Y - \mu a^{[r-1]} W^{[r]}||_2^2 + \sum_{j=1}^{n^{[r-1]}} (\sigma ||a_j^{[r-1]}||_2 W_j^{[r]})^2.$$

Thus, we conclude that Eq. 21 decomposes into Eq. 22, since

$$||\Gamma W^{[r]}||_2^2 = ||\sigma G W^{[r]}||_2^2 = \sum_{j=1}^{n^{[r-1]}} (\sigma ||a_j^{[r-1]}||_2 W_j^{[r]})^2,$$

where the inverse of the column normalization matrix $G = \text{diag}(||a_1^{[r-1]}||_2, \ldots, ||a_{n^{[r-1]}}^{[r-1]}||_2)$. Moreover, we observe that the expected loss under dropout decomposes into a ridge regression problem

$$\mathbb{E}[J(W^{[1]}, \cdots, W^{[r]})] = ||Y - \mu a^{[r-1]} W^{[r]}||_2^2 + \sigma^2 ||W^{[r]}||_2^2,$$

when we normalize the columns of the penultimate layer activations such that the Tikhonov matrix $\Gamma = \sigma G = \sigma \mathbb{I}$, i.e., $||a_j^{[r-1]}||_2 = 1$ for $j = 1, \ldots, n^{[r-1]}$.

### C.3.1 Derivatives of the expected loss under dropout with respect to the last layer weights

From Eq. 22, we have that

$$\frac{1}{2}\nabla_{W^{[r]}}(\mathbb{E}[J(W^{[1]}, \cdots, W^{[r]})]) = \frac{1}{2}\nabla_{W^{[r]}}(||Y - \mu a^{[r-1]}W^{[r]}||_2^2) + \frac{1}{2}\nabla_{W^{[r]}}(||\Gamma W^{[r]}||_2^2),$$

where the decomposition enables us to compute the gradient of the expected loss under dropout with respect to the last layer weights. In particular, we have that

$$\frac{1}{2}\nabla_{W^{[r]}}(\mathbb{E}[J(W^{[1]}, \cdots, W^{[r]})]) = \mu(a^{[r-1]})^t(\mu a^{[r-1]}W^{[r]} - Y) + \Gamma^t\Gamma W^{[r]},$$

which is equal to

$$\mu(a^{[r-1]})^t(\mu a^{[r-1]}W^{[r]} - Y) + \sigma^2 G^t G W^{[r]},$$

where the square of the mean in the term $\mu^2(a^{[r-1]})^t a^{[r-1]}W^{[r]}$ implies that we must reformulate the gradient of the expected loss under dropout with respect to the last layer weights in terms of the re-scaled penultimate layer activations.

Consequently, if we reformulate, then we have that

$$\frac{1}{2}\nabla_{W^{[r]}}(\mathbb{E}[J(W^{[1]}, \cdots, W^{[r]})]) = (\tilde{a}^{[r-1]})^t(\tilde{a}^{[r-1]}W^{[r]} - Y) + (\frac{\sigma}{\mu})^2\tilde{G}^t\tilde{G}W^{[r]}.$$

Moreover, if we column normalize and reformulate, then we have that

$$\frac{1}{2}\nabla_{W^{[r]}}(\mathbb{E}[J(W^{[1]}, \cdots, W^{[r]})]) = (\tilde{a}^{[r-1]})^t(\tilde{a}^{[r-1]}W^{[r]} - Y) + \sigma^2 W^{[r]}.$$

Correspondingly, if we reformulate, then we have that the Hessian of the expected loss under dropout with respect to the last layer weights is

$$\frac{1}{2}D(\nabla_{W^{[r]}}(\mathbb{E}[J(W^{[1]}, \cdots, W^{[r]})])) = (\tilde{a}^{[r-1]})^t\tilde{a}^{[r-1]} + (\frac{\sigma}{\mu})^2\tilde{G}^t\tilde{G},$$

Otherwise, if we column normalize and reformulate, then we have that the Hessian of the expected loss under dropout with respect to the last layer weights is

$$\frac{1}{2}D(\nabla_{W^{[r]}}(\mathbb{E}[J(W^{[1]}, \cdots, W^{[r]})])) = (\tilde{a}^{[r-1]})^t\tilde{a}^{[r-1]} + \sigma^2\mathbb{I}.$$

Therefore, we have that the dropout-induced geometry retains the connections that we found in section C.2.1.

### C.3.2 Derivatives of the expected loss under dropout with respect to the re-scaled last layer weights

Next, we want to show that the work in section C.3.1 persists, when we compute the derivatives of the expected loss under dropout with respect to the re-scaled last layer weights. From Eq. 22, we have that

$$\frac{1}{2}\nabla_{\tilde{W}^{[r]}}(\mathbb{E}[J(W^{[1]}, \cdots, W^{[r]})]) = \frac{1}{2}\nabla_{\tilde{W}^{[r]}}(||Y - \mu a^{[r-1]}W^{[r]}||_2^2) + \frac{1}{2}\nabla_{\tilde{W}^{[r]}}(||\Gamma W^{[r]}||_2^2),$$

where the decomposition enables us to compute the gradient of the expected loss under dropout with respect to the re-scaled last layer weights, when we reformulate the decomposition in terms of the re-scaled last layer weights.

Namely, we have that

$$\frac{1}{2}\nabla_{\tilde{W}^{[r]}}(\mathbb{E}[J(W^{[1]}, \cdots, W^{[r]})]) = (a^{[r-1]})^t(a^{[r-1]}\tilde{W}^{[r]} - Y) + \frac{1}{\mu^2}\Gamma^t\Gamma\tilde{W}^{[r]},$$

which is equal to

$$(a^{[r-1]})^t(a^{[r-1]}\tilde{W}^{[r]} - Y) + (\frac{\sigma}{\mu})^2 G^t G\tilde{W}^{[r]}.$$

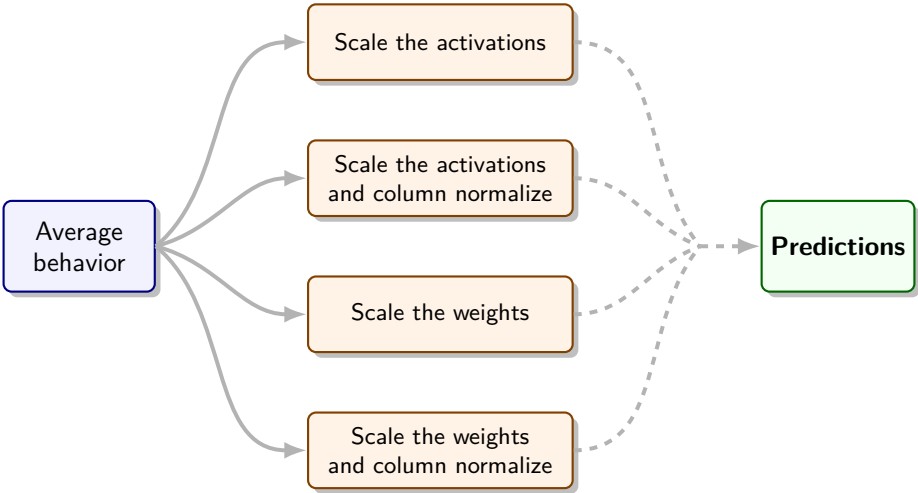

Figure 5: We reformulate the average behavior of a fully-connected neural network with dropout in the last layer in four different ways in order to disambiguate the dropout-induced regularization and optimization. Then, we leave open the question of whether or not the reformulations result in the same predictions at test time.

Moreover, if we column normalize and reformulate, then we have that

$$\frac{1}{2}\nabla_{\tilde{W}^{[r]}}(\mathbb{E}[J(W^{[1]}, \cdots, W^{[r]})]) = (a^{[r-1]})^t(a^{[r-1]}\tilde{W}^{[r]} - Y) + (\frac{\sigma}{\mu})^2\tilde{W}^{[r]}.$$

Correspondingly, if we reformulate, then we have that the Hessian of the expected loss under dropout with respect to the re-scaled last layer weights is

$$\frac{1}{2}D(\nabla_{\tilde{W}^{[r]}}(\mathbb{E}[J(W^{[1]}, \cdots, W^{[r]})])) = (a^{[r-1]})^t a^{[r-1]} + (\frac{\sigma}{\mu})^2 G^t G,$$

Otherwise, if we column normalize and reformulate, then we have that the Hessian of the expected loss under dropout with respect to the re-scaled last layer weights is

$$\frac{1}{2}D(\nabla_{\tilde{W}^{[r]}}(\mathbb{E}[J(W^{[1]}, \cdots, W^{[r]})])) = (a^{[r-1]})^t a^{[r-1]} + (\frac{\sigma}{\mu})^2 \mathbb{I}.$$

Therefore, we conclude that the work in section C.3.1 persists, when we compute the derivatives of the expected loss under dropout with respect to the re-scaled last layer weights. Moreover, we have that the dropout-induced geometry retains the connections that we found in section C.2.2.

## C.4 Dual interpretations

From section C.2, we found that there exist multiple interpretations of the average behavior of a fully-connected neural network with dropout in the last layer such that each interpretation details the regularization strength and optimization problem to solve based on the ways in which a practitioner is able to scale a quantity and choose whether or not to column normalize the penultimate layer activations. However, we do not possess a closed form solution in this context of machine learning, so we leave open the question of whether or not the reformulations result in the same predictions at test time; see Figure 5.

## C.5 Penalty term under dropout

Nevertheless, we are able to explore the penalty term under dropout and analyze the ways in which it differs from the base penalty term, which provides insights into the train time behavior of a fully-connected neural

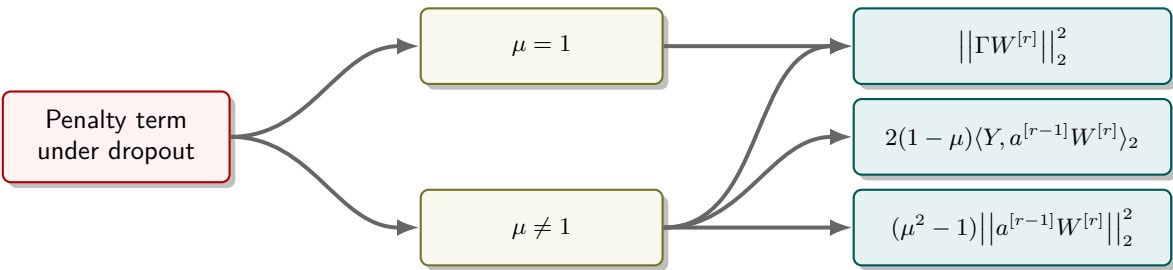

Figure 6: In the case of $\mu = 1$, the penalty term under dropout corresponds to Tikhonov regularization, where the Tikhonov matrix takes the form that we observe in Eq. 22. Otherwise, the penalty term under dropout generalizes this particular form of Tikhonov regularization with the addition of two terms, which depend on the overlap between the targets and predictions at train time as well as the size of the predictions at train time.

network with dropout in the last layer. To begin, we define the base penalty term to be

$$\left|\left|\Gamma W^{[r]}\right|\right|_2^2,$$

which is the data-dependent Tikhonov regularization term in Eq. 22. Then, we compute the penalty term under dropout

$$\mathbb{E}[J(W^{[1]}, \cdots, W^{[r]})] - J(W^{[1]}, \cdots, W^{[r]}) = 2(1-\mu)\langle Y, a^{[r-1]}W^{[r]}\rangle_2 + (\mu^2 - 1)\left|\left|a^{[r-1]}W^{[r]}\right|\right|_2^2 + \left|\left|\Gamma W^{[r]}\right|\right|_2^2,$$
$$(27)$$

which is the difference between the decomposition in Eq. 22 and the squared error loss function in Eq. 20.

From the right-hand side of Eq. 27, we have that the penalty term under dropout includes two regularization terms that go beyond the base regularization term.

- In particular, the first term corresponds to a scalar multiple of the inner product between the targets and predictions at train time $2(1-\mu)\langle Y, a^{[r-1]}W^{[r]}\rangle_2$, which shows that the penalty term under dropout depends on the overlap between the targets and predictions at train time.

- Then, the second term corresponds to a scalar multiple of the square of the norm of the predictions at train time $(\mu^2 - 1)\left|\left|a^{[r-1]}W^{[r]}\right|\right|_2^2$, which shows that the penalty term under dropout depends on the size of the predictions at train time.

Interestingly, we observe that the two additional terms on the right-hand side of Eq. 27 vanish, when a practitioner picks a dropout distribution such that $\mu = 1$, i.e., inverted dropout or Gaussian dropout. Otherwise, the entire penalty term under dropout remains intact, when a practitioner picks a dropout distribution such that $\mu \neq 1$, i.e., a Bernoulli distribution with $\mu = p \in (0, 1)$; see Figure 6.

## D  Generalized linear models

Finally, we examine generalized linear models. Here,

$$J(\theta) = -\sum_{i=1}^{m} \ln(h(Y_i)) + Y_i(X\theta)_i - A((X\theta)_i) \qquad (28)$$

denotes the generalized linear model loss function, which is the negative log-likelihood of the conditional exponential family distribution. Specifically, the term $h(Y_i)$ is the underlying measure, which is a function of the label $Y_i$, where $i = 1, \ldots, m$. Next, the term $Y_i(X\theta)_i$ is the product of the sufficient statistic $T(Y_i) = Y_i$ and natural parameter $(X\theta)_i$, where $X$ is the design matrix with $m$ examples and $n$ features, $\theta$ is the collection of $n$ parameters, and $i = 1, \ldots, m$. Then, the term $A((X\theta)_i)$ is the log-partition function, where $i = 1, \ldots, m$. Overall, the objective is to find the collection of parameters, which minimize the loss function.

### D.1 Generalized linear models with dropout

To study generalized linear models with dropout, we draw independent and identically distributed random variables $d_j \overset{\text{iid}}{\sim} \mathfrak{D}$ from a probability distribution that has finite but non-zero mean $\mathbb{E}[d_j] = \mu$ and finite variance $\text{Var}(d_j) = \sigma^2$, where $j = 1, \ldots, n$. Next, we construct a dropout mask $\delta = \text{diag}(d_1, \ldots, d_n)$, which is a diagonal matrix with the dropout random variables on the diagonal. Then, we form the composite dropped-out product

$$X\delta\theta = (X\delta)\theta = X(\delta\theta),$$

where $X\delta$ and $\delta\theta$ correspond to the dropped-out design matrix and parameters, respectively. Moreover, the term $(X\delta)_i$ corresponds to a dropped-out example for $i = 1, \ldots, m$.

### D.2 Expected loss under dropout

Accordingly, we form the loss under dropout

$$J(\theta) = -\sum_{i=1}^{m} \ln(h(Y_i)) + Y_i(X\delta\theta)_i - A((X\delta\theta)_i).$$

Next, we consider the expected loss under dropout

$$\mathbb{E}[J(\theta)] = -\sum_{i=1}^{m} \mathbb{E}[\ln(h(Y_i)) + Y_i(X\delta\theta)_i - A((X\delta\theta)_i)], \tag{29}$$

since this expectation value enables us to analyze the average behavior of generalized linear models with dropout. Then, we have that Eq. 29 decomposes into a sum of a loss term and a data-dependent regularization term

$$\mathbb{E}[J(\theta)] = -\sum_{i=1}^{m} \ln(h(Y_i)) + Y_i(\mu X\theta)_i - A((\mu X\theta)_i) - \mathbb{E}[A((X\delta\theta)_i)] + A((\mu X\theta)_i), \tag{30}$$

where the loss term is

$$J(\theta) = -\sum_{i=1}^{m} \ln(h(Y_i)) + Y_i(\mu X\theta)_i - A((\mu X\theta)_i),$$

and the regularization term is

$$R(\theta) = \sum_{i=1}^{m} \mathbb{E}[A((X\delta\theta)_i)] - A((\mu X\theta)_i). \tag{31}$$

Additionally, we note that Eq. 30 generalizes the decomposition and Eq. 31 generalizes the regularization term in the seminal work of Wager et al. (2013); see the details in section D.4.

#### D.2.1 Quadratic approximation of the regularization term under dropout

Similar to the works of Webb (1994), Bishop (1995), and Wager et al. (2013), we derive a quadratic approximation of Eq. 31, since this enables us to reason about the average behavior of generalized linear models with dropout in a way that resembles sections 2.1.1, B.1.1, and C.1.1. To this end, we define the term $Z_i = (\mu X\theta)_i$ and difference $\epsilon_i = (X\delta\theta)_i - (\mu X\theta)_i$ such that $Z_i + \epsilon_i = (X\delta\theta)_i$ for $i = 1, \ldots, m$. Subsequently, we define a diagonal matrix $\text{diag}(\theta_1, \ldots, \theta_n)$, which has the parameters on the diagonal. Then, we have that the quadratic approximation of the regularization term under dropout is

$$R(\theta) = \frac{1}{2} \left\| \Gamma \, \text{diag}(\theta_1, \ldots, \theta_n) \right\|_F^2, \tag{32}$$

where the Tikhonov matrix is a scalar multiple $\sigma$ of a matrix G such that

$$G = \text{diag}\left(\sqrt{A''(Z_1)}, \ldots, \sqrt{A''(Z_m)}\right) X$$

and $A''(Z_i)$ is the variance function for $i = 1, \ldots, m$. Moreover, we note that Eq. 32 generalizes the quadratic approximation of the regularization term under dropout in the work of Wager et al. (2013); see the details in section D.4.1.

### D.2.2 Quadratic approximation of the expected loss under dropout

From there, we make a quadratic approximation of the expected loss under dropout

$$\mathbb{E}[J(\theta)] \approx -\sum_{i=1}^{m} \ln(h(Y_i)) + Y_i(\mu X\theta)_i - A((\mu X\theta)_i) + \frac{1}{2}\big|\big|\Gamma\,\mathrm{diag}\,(\theta_1,\ldots,\theta_n)\big|\big|_F^2, \tag{33}$$

where we retain the loss term in Eq. 30 and approximate Eq. 31 with Eq. 32. Thereupon, we observe that Eq. 33 is many-to-one in terms of the original dropped-out quantity. The implication of this property of the approximate average behavior is that our interpretation of the dropout-induced regularization and optimization is ambiguous, when we think from the perspective of Eq. 33.

### D.3 Reformulation of the decomposition

Correspondingly, we reformulate Eq. 33 in terms of the quantity to scale and choice of whether or not to column normalize the weighted design matrix, which is consistent with our approach in section 2.2. That is, we consider a practitioner, who chooses to scale the examples in accord with standard dropout in section D.3.1 and the parameters in section D.3.2.

### D.3.1 Scale the examples

To carry this out, we begin with a practitioner, who chooses to scale the examples in accord with standard dropout. Next, we map the original design matrix to the re-scaled design matrix $\tilde{X} = \mu X = \mathbb{E}[X\delta]$. Then, we have that the approximate decomposition becomes

$$\begin{aligned}
\mathbb{E}[J(\theta)] \approx\ & -\sum_{i=1}^{m} \ln(h(Y_i)) + Y_i(\tilde{X}\theta)_i - A((\tilde{X}\theta)_i) \\
& + \frac{1}{2}\big(\frac{\sigma}{\mu}\big)^2 \big|\big|\,\mathrm{diag}\,(\sqrt{A''((\tilde{X}\theta)_1)},\ldots,\sqrt{A''((\tilde{X}\theta)_m)})\tilde{X}\,\mathrm{diag}\,(\theta_1,\ldots,\theta_n)\big|\big|_F^2.
\end{aligned} \tag{34}$$

Accordingly, we find that the regularization strength equals the square of the coefficient of variation $\lambda = \big(\frac{\sigma}{\mu}\big)^2$, when we think about the nature of dropout from the perspective of Eq. 34. [11] Moreover, we observe that the optimization problem of interest is in terms of the original parameters and there is symmetry in the appearance of the re-scaled design matrix in the loss term and regularization term.

Similarly, we consider a practitioner, who chooses to scale the examples and column normalize the weighted design matrix such that $||(\mathrm{diag}\,(\sqrt{A''(Z_1)},\ldots,\sqrt{A''(Z_m)})X)_j||_2 = 1$ for $j = 1,\ldots,n$. Then, we have that the approximate decomposition becomes

$$\mathbb{E}[J(\theta)] \approx -\sum_{i=1}^{m} \ln(h(Y_i)) + Y_i(\tilde{X}\theta)_i - A((\tilde{X}\theta)_i) + \frac{1}{2}\sigma^2\big|\big|\theta\big|\big|_2^2. \tag{35}$$

Consequently, we find that the regularization strength equals the variance $\lambda = \sigma^2$, when we think about the nature of dropout from the perspective of Eq. 35. Moreover, we observe that the optimization problem of interest is in terms of the original parameters and there is asymmetry in the appearance of the re-scaled design matrix in the loss term and regularization term.

Additionally, we reformulate the gradient and Hessian of the quadratic approximation of the regularization term under dropout with respect to the parameters in section D.4.2. [12] These computations show that the connections in the dropout-induced regularization persist in the geometry.

---

[11] We do not include the prefactor of $\frac{1}{2}$ from the quadratic approximation, when we specify the regularization strength.

[12] We focus on these derivatives, since the reformulation of the gradient and Hessian of the quadratic approximation of the expected loss under dropout follows in a straightforward way from this work.

### D.3.2 Scale the parameters

Ensuingly, we consider a practitioner, who chooses to scale the parameters. Next, we map the original parameters to the re-scaled parameters $\tilde{\theta} = \mu\theta = \mathbb{E}[\delta\theta]$. Then, we have that the approximate decomposition becomes

$$
\begin{aligned}
J(\tilde{\theta}) = & -\sum_{i=1}^{m} \ln(h(Y_i)) + Y_i(X\tilde{\theta})_i - A((X\tilde{\theta})_i) \\
& + \frac{1}{2}(\frac{\sigma}{\mu})^2 \big|\big| \operatorname{diag}\left(\sqrt{A''((X\tilde{\theta})_1)}, \ldots, \sqrt{A''((X\tilde{\theta})_m)}\right) X \operatorname{diag}\left(\tilde{\theta}_1, \ldots, \tilde{\theta}_n\right) \big|\big|_F^2,
\end{aligned}
\tag{36}
$$

which is a generalized linear model loss function of the re-scaled parameters with a Tikhonov regularization term.

Accordingly, we find that the regularization strength equals the square of the coefficient of variation $\lambda = (\frac{\sigma}{\mu})^2$, when we think about the nature of dropout from the perspective of Eq. 36. Moreover, we observe that the optimization problem of interest is in terms of the re-scaled parameters and there is symmetry in the appearance of the re-scaled parameters in the loss term and regularization term.

Similarly, we consider a practitioner, who chooses to scale the parameters and column normalize the weighted design matrix. Then, we have that the approximate decomposition becomes

$$
J(\tilde{\theta}) = -\sum_{i=1}^{m} \ln(h(Y_i)) + Y_i(X\tilde{\theta})_i - A((X\tilde{\theta})_i) + \frac{1}{2}(\frac{\sigma}{\mu})^2 \big|\big|\tilde{\theta}\big|\big|_2^2,
\tag{37}
$$

which is a generalized linear model loss function of the re-scaled parameters with a ridge regularization term.

Consequently, we find that the regularization strength equals the square of the coefficient of variation $\lambda = (\frac{\sigma}{\mu})^2$, when we think about the nature of dropout from the perspective of Eq. 37. Moreover, we observe that the optimization problem of interest is in terms of the re-scaled parameters and there is symmetry in the appearance of the re-scaled parameters in the loss term and regularization term.

Furthermore, we reformulate the gradient and Hessian of the quadratic approximation of the regularization term under dropout with respect to the re-scaled parameters in section D.4.3. These computations show that the connections in the dropout-induced regularization persist in the geometry.

## D.4 Derivation of the decomposition

To derive Eq. 30, we begin with Eq. 29

$$
\mathbb{E}[J(\theta)] = -\sum_{i=1}^{m} \mathbb{E}[\ln(h(Y_i)) + Y_i(X\delta\theta)_i - A((X\delta\theta)_i)],
$$

which simplifies to

$$
-\sum_{i=1}^{m} \ln(h(Y_i)) + Y_i(\mu X\theta)_i - \mathbb{E}[A((X\delta\theta)_i)].
$$

Next, we add zero $0 = \sum_{i=1}^{m} A((\mu X\theta)_i) - A((\mu X\theta)_i)$ to the sum and rearrange the terms to obtain

$$
-\sum_{i=1}^{m} \ln(h(Y_i)) + Y_i(\mu X\theta)_i - A((\mu X\theta)_i) - \mathbb{E}[A((X\delta\theta)_i)] + A((\mu X\theta)_i).
$$

Thus, we conclude that Eq. 29 decomposes into Eq. 30, since the loss term is

$$
J(\theta) = -\sum_{i=1}^{m} \ln(h(Y_i)) + Y_i(\mu X\theta)_i - A((\mu X\theta)_i),
$$

and the regularization term is

$$R(\theta) = \sum_{i=1}^{m} \mathbb{E}[A((X\delta\theta)_i)] - A((\mu X\theta)_i).$$

Moreover, we observe that the loss term reduces to Eq. 28 and the regularization term reduces to

$$R(\theta) = \sum_{i=1}^{m} \mathbb{E}[A((X\delta\theta)_i)] - A((X\theta)_i),$$

when we set the mean of the dropout distribution such that $\mu = 1$. Therefore, we show that Eq. 30 generalizes the decomposition and Eq. 31 generalizes the regularization term in the work of Wager et al. (2013). [13]

### D.4.1 Derivation of the quadratic approximation of the regularization term under dropout

Ensuingly, we derive Eq. 32, which is a quadratic approximation of Eq. 31. To do so, we define $Z_i = (\mu X\theta)_i$ and $\epsilon_i = (X\delta\theta)_i - (\mu X\theta)_i$ such that $Z_i + \epsilon_i = (X\delta\theta)_i$ for $i = 1, \ldots, m$. Next, we consider

$$A(Z_i + \epsilon_i) = A(Z_i) + A'(Z_i)\epsilon_i + \frac{1}{2}A''(Z_i)\epsilon_i^2 + R_2(Z_i, \epsilon_i),$$

which is a second-order Taylor expansion of the log-partition function at $Z_i$ such that $R_2(Z_i, \epsilon_i)$ is the remainder term and $i = 1, \ldots, m$. Then, we rearrange terms in the second-order Taylor expansion to obtain

$$A(Z_i + \epsilon_i) - A(Z_i) = A'(Z_i)\epsilon_i + \frac{1}{2}A''(Z_i)\epsilon_i^2 + R_2(Z_i, \epsilon_i),$$

where $i = 1, \ldots, m$.

Correspondingly, we observe that Eq. 31 equals

$$R(\theta) = \sum_{i=1}^{m} \mathbb{E}[A(Z_i + \epsilon_i)] - A(Z_i) = \sum_{i=1}^{m} \mathbb{E}[A(Z_i + \epsilon_i) - A(Z_i)],$$

which approximates to

$$\sum_{i=1}^{m} \mathbb{E}[A'(Z_i)\epsilon_i + \frac{1}{2}A''(Z_i)\epsilon_i^2] = \sum_{i=1}^{m} A'(Z_i)\mathbb{E}[\epsilon_i] + \frac{1}{2}A''(Z_i)\mathbb{E}[\epsilon_i^2],$$

when we discard the remainder term. Then, we have that the approximant reduces to

$$\sum_{i=1}^{m} \frac{1}{2}A''(Z_i)\operatorname{Var}(\epsilon_i) = \sum_{i=1}^{m} \frac{1}{2}A''(Z_i)\operatorname{Var}(Z_i + \epsilon_i) = \frac{1}{2}\sum_{i=1}^{m}\sum_{j=1}^{n}(\sigma\sqrt{A''(Z_i)}X_{ij}\theta_j)^2,$$

since the expectation vanishes $\mathbb{E}[\epsilon_i] = (\mu X\theta)_i - (\mu X\theta)_i = 0$, $Z_i$ is a constant, the variance equals

$$\operatorname{Var}(Z_i + \epsilon_i) = \sigma^2 \sum_{j=1}^{n} X_{ij}^2 \theta_j^2,$$

and $i = 1, \ldots, m$. Finally, we obtain Eq. 32, since

$$\left\|\Gamma \operatorname{diag}(\theta_1, \ldots, \theta_n)\right\|_F^2 = \left\|\sigma G \operatorname{diag}(\theta_1, \ldots, \theta_n)\right\|_F^2 = \sum_{i=1}^{m}\sum_{j=1}^{n}(\sigma\sqrt{A''(Z_i)}X_{ij}\theta_j)^2,$$

---

[13]To check this, we note that the middle equality in equation (4) of Wager et al. (2013) requires a subscript on the label and the inclusion of the underlying measure.

where $G = \text{diag}\,(\sqrt{A''(Z_1)}, \ldots, \sqrt{A''(Z_m)})X$ is the weighted design matrix and $\text{diag}\,(\theta_1, \ldots, \theta_n)$ is a diagonal matrix with the parameters on the diagonal.

Furthermore, we observe that the quadratic approximation of the regularization term under dropout reduces to

$$R(\theta) = \frac{1}{2}\sum_{i=1}^{m}\sum_{j=1}^{n}(\sigma\sqrt{A''((X\theta)_i)}X_{ij}\theta_j)^2,$$

when we set the mean of the dropout distribution such that $\mu = 1$. Therefore, we show that Eq. 32 generalizes the quadratic approximation of the regularization term under dropout in the work of Wager et al. (2013). [14]

### D.4.2 Derivatives of the quadratic approximation of the regularization term under dropout with respect to the parameters

From Eq. 32, we have that

$$\nabla_\theta(R(\theta)) = \frac{1}{2}\nabla_\theta(\left\|\Gamma\,\text{diag}\,(\theta_1, \ldots, \theta_n)\right\|_F^2),$$

which enables us to compute the gradient of the quadratic approximation of the regularization term under dropout with respect to the parameters. In particular, we have that

$$\nabla_\theta(R(\theta)) = \frac{\sigma^2\mu}{2}(\theta \odot \theta)(X \odot X)^t\,\text{diag}\,(A'''(Z_1), \ldots, A'''(Z_m))X$$
$$+ \sigma^2\theta \odot (A''(Z_1), \ldots, A''(Z_m))(X \odot X),$$

where $A''(Z_i)$ is the variance function, $A'''(Z_i)$ is the third derivative of the log-partition function at the natural parameter, and $i = 1, \ldots, m$. Then, the terms imply that we must reformulate the gradient of the quadratic approximation of the regularization term under dropout with respect to the parameters in terms of the re-scaled design matrix.

Consequently, if we reformulate, then we have that

$$\nabla_\theta(R(\theta)) = \frac{1}{2}(\frac{\sigma}{\mu})^2(\theta \odot \theta)(\tilde{X} \odot \tilde{X})^t\,\text{diag}\,(A'''((\tilde{X}\theta)_1), \ldots, A'''((\tilde{X}\theta)_m))\tilde{X}$$
$$+ (\frac{\sigma}{\mu})^2\theta \odot (A''((\tilde{X}\theta)_1), \ldots, A''((\tilde{X}\theta)_m))(\tilde{X} \odot \tilde{X}).$$

Moreover, if we column normalize and reformulate, then we have that

$$\nabla_\theta(R(\theta)) = \sigma^2\theta.$$

Correspondingly, if we reformulate, then we have that the Hessian of the quadratic approximation of the regularization term under dropout with respect to the parameters is

$$D(\nabla_\theta(R(\theta))) = (\frac{\sigma}{\mu})^2\tilde{X}^t\,\text{diag}\,(A'''((\tilde{X}\theta)_1), \ldots, A'''((\tilde{X}\theta)_m))(\tilde{X} \odot \tilde{X})\,\text{diag}\,(\theta_1, \ldots, \theta_n)$$
$$+ \frac{1}{2}(\frac{\sigma}{\mu})^2\tilde{X}^t\,\text{diag}\,((A''''((\tilde{X}\theta)_1), \ldots, A''''((\tilde{X}\theta)_m)) \odot (\tilde{X} \odot \tilde{X})(\theta \odot \theta))\tilde{X}$$
$$+ (\frac{\sigma}{\mu})^2\,\text{diag}\,((A''((\tilde{X}\theta)_1), \ldots, A''((\tilde{X}\theta)_m))(\tilde{X} \odot \tilde{X}))$$
$$+ (\frac{\sigma}{\mu})^2\,\text{diag}\,(\theta_1, \ldots, \theta_n)(\tilde{X} \odot \tilde{X})^t\,\text{diag}\,(A'''((\tilde{X}\theta)_1), \ldots, A'''((\tilde{X}\theta)_m))\tilde{X}.$$

Otherwise, if we column normalize and reformulate, then we have that the Hessian of the quadratic approximation of the regularization term under dropout with respect to the parameters is

$$D(\nabla_\theta(R(\theta))) = \sigma^2\mathbb{I}.$$

Therefore, we have that the dropout-induced geometry retains the connections that we found in section D.2.1.

---

[14]To check this, we note that equation (9) of Wager et al. (2013) requires removal of a multiplicative factor of $\frac{1}{2}$.

### D.4.3 Derivatives of the quadratic approximation of the regularization term under dropout with respect to the re-scaled parameters

Next, we want to show that the work in section D.4.2 persists, when we compute the derivatives of the quadratic approximation of the regularization term under dropout with respect to the re-scaled parameters. From Eq. 32, we have that

$$\nabla_{\tilde{\theta}}(R(\theta)) = \frac{1}{2}\nabla_{\tilde{\theta}}(\left|\left|\Gamma \operatorname{diag}(\theta_1, \ldots, \theta_n)\right|\right|_F^2),$$

which enables us to compute the gradient of the quadratic approximation of the regularization term under dropout with respect to the re-scaled parameters, when we reformulate the quadratic approximation of the regularization term under dropout in terms of the re-scaled parameters.

Namely, we have that

$$\nabla_{\tilde{\theta}}(R(\theta)) = \frac{1}{2}(\frac{\sigma}{\mu})^2(\tilde{\theta} \odot \tilde{\theta})(X \odot X)^t \operatorname{diag}(A'''((X\tilde{\theta})_1), \ldots, A'''((X\tilde{\theta})_m))X$$
$$+ (\frac{\sigma}{\mu})^2\tilde{\theta} \odot (A''((X\tilde{\theta})_1), \ldots, A''((X\tilde{\theta})_m))(X \odot X).$$

Moreover, if we column normalize and reformulate, then we have that

$$\nabla_{\tilde{\theta}}(R(\theta)) = (\frac{\sigma}{\mu})^2\tilde{\theta}.$$

Correspondingly, if we reformulate, then we have that the Hessian of the quadratic approximation of the regularization term under dropout with respect to the re-scaled parameters is

$$D(\nabla_{\tilde{\theta}}(R(\theta))) = (\frac{\sigma}{\mu})^2 X^t \operatorname{diag}(A'''((X\tilde{\theta})_1), \ldots, A'''((X\tilde{\theta})_m))(X \odot X) \operatorname{diag}(\tilde{\theta}_1, \ldots, \tilde{\theta}_n)$$
$$+ \frac{1}{2}(\frac{\sigma}{\mu})^2 X^t \operatorname{diag}((A''''((X\tilde{\theta})_1), \ldots, A''''((X\tilde{\theta})_m)) \odot (X \odot X)(\tilde{\theta} \odot \tilde{\theta}))X$$
$$+ (\frac{\sigma}{\mu})^2 \operatorname{diag}((A''((X\tilde{\theta})_1), \ldots, A''((X\tilde{\theta})_m))(X \odot X))$$
$$+ (\frac{\sigma}{\mu})^2 \operatorname{diag}(\tilde{\theta}_1, \ldots, \tilde{\theta}_n)(X \odot X)^t \operatorname{diag}(A'''((X\tilde{\theta})_1), \ldots, A'''((X\tilde{\theta})_m))X.$$

Otherwise, if we column normalize and reformulate, then we have that the Hessian of the quadratic approximation of the regularization term under dropout with respect to the re-scaled parameters is

$$D(\nabla_{\tilde{\theta}}(R(\theta))) = (\frac{\sigma}{\mu})^2\mathbb{I}.$$

Therefore, we conclude that the work in section D.4.2 persists, when we compute the derivatives of the quadratic approximation of the regularization term under dropout with respect to the re-scaled parameters. Moreover, we have that the dropout-induced geometry retains the connections that we found in section D.3.2.

### D.5 Dual interpretations

From section D.3, we found that there exist multiple interpretations of the approximate average behavior of generalized linear models with dropout such that each interpretation details the regularization strength and optimization problem to solve based on the ways in which a practitioner is able to scale a quantity and choose whether or not to column normalize the weighted design matrix. However, we do not possess a closed form solution in this context of machine learning, so we leave open the question of whether or not the reformulations result in the same predictions at test time; see Figure 7.

### D.6 Penalty term under dropout

Nevertheless, we are able to explore the penalty term under dropout and analyze the ways in which it differs from the base penalty term, which provides insights into the train time behavior of generalized linear models

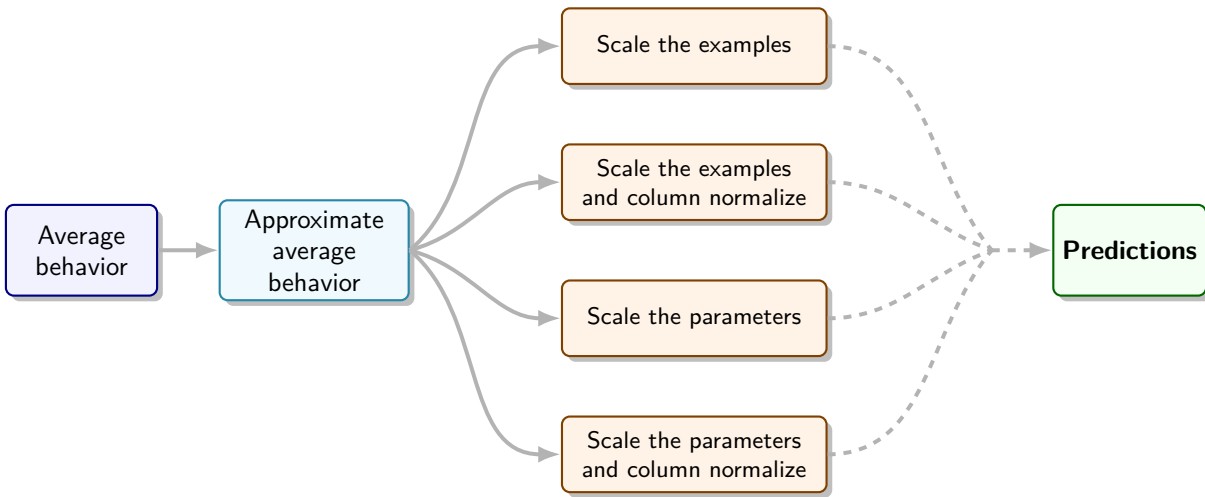

Figure 7: We reformulate the approximate average behavior of generalized linear models with dropout in four different ways in order to disambiguate the dropout-induced regularization and optimization. Then, we leave open the question of whether or not the reformulations result in the same predictions at test time.

with dropout. To begin, we define the base penalty term to be

$$R(\theta) = \sum_{i=1}^{m} \mathbb{E}[A((X\delta\theta)_i)] - A((\mu X\theta)_i).$$

which is the data-dependent regularization term in Eq. 30. Then, we compute the penalty term under dropout

$$\mathbb{E}[J(\theta)] - J(\theta) = (1 - \mu)\langle Y, X\theta\rangle_2 + \sum_{i=1}^{m} A((\mu X\theta)_i) - A((X\theta)_i) + R(\theta), \tag{38}$$

which is the difference between the decomposition in Eq. 30 and the generalized linear model loss function in Eq. 28.

From the right-hand side of Eq. 38, we have that the penalty term under dropout includes two regularization terms that go beyond the base regularization term.

- In particular, the first term corresponds to a scalar multiple of the inner product between the targets and predictions at train time $(1 - \mu)\langle Y, X\theta\rangle_2$, which shows that the penalty term under dropout depends on the overlap between the targets and predictions at train time.

- Then, the second term corresponds to the difference between the log-partition function at the expected dropped-out prediction at train time and the log-partition function at the prediction at train time $\sum_{i=1}^{m} A((\mu X\theta)_i) - A((X\theta)_i)$, which shows that the penalty term under dropout depends on the predictions at train time.

Interestingly, we observe that the two additional terms on the right-hand side of Eq. 38 vanish, when a practitioner picks a dropout distribution such that $\mu = 1$, i.e., inverted dropout or Gaussian dropout. Otherwise, the entire penalty term under dropout remains intact, when a practitioner picks a dropout distribution such that $\mu \neq 1$, i.e., a Bernoulli distribution with $\mu = p \in (0, 1)$; see Figure 8.

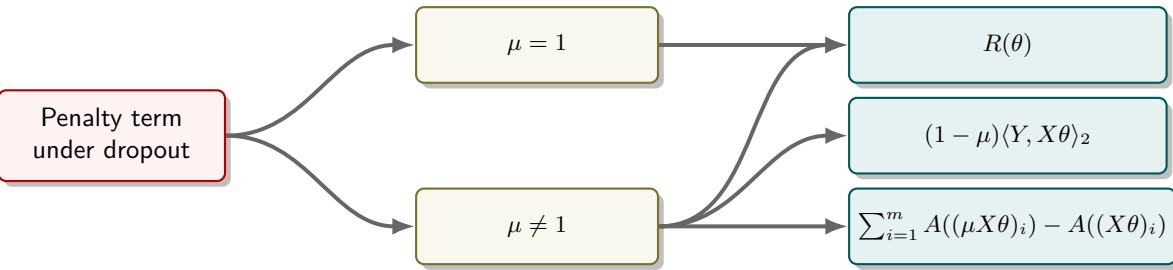

Figure 8: In the case of $\mu = 1$, the penalty term under dropout corresponds to the regularization term in Eq. 31. Otherwise, the penalty term under dropout generalizes this particular regularization term with the addition of two terms, which depend on the overlap between the targets and predictions at train time as well as the predictions at train time.

