# OpenReview forum: "A new perspective on the nature of dropout"
_TMLR — Rejected by TMLR_

### Review · Reviewer_ZBgj · 2026-05-17

**Summary Of Contributions:**

This paper studies the average behavior of dropout in several contexts including linear regression, generalized linear models. and neural networks, among others, and showed that the average behavior does not distinguish whether the dropout mask is interpreted as acting on the data, parameters, activations, or weight. The paper then proposes an “operational” interpretation based on what a practitioner can scale or normalize, and argues that different operational reformulations lead to different apparent regularization strengths. In linear regression, the paper further shows that these different reformulations nevertheless yield the same test-time predictions

Strengths:
1. The observation that the expected dropout loss does not intrinsically encode whether dropout is viewed as acting on features, parameters, activations, or weights is reasonable and interesting.
2. The linear regression derivations are mostly transparent and useful.
3. The paper correctly highlights the special role of inverted dropout, where in practice, dropout is usually scaled so that \mu=1, and this paper explains, to a certain degree, that why prior analyses often did not need to distinguish the more general case.

Weakness:
1. The main contribution is largely interpretive rather than genuine new theoretical results. Moreover, it is only well-supported on a narrow setting, linear regression. For the nonlinear/nonconvex cases, the paper does not prove equivalence of test-time outputs and explicitly leaves this as an open question. The authors should either narrow the claim, or strengthen the supporting evidence.

2. There might be a serious sign error for the GLM section. For negative log-likelihood, I thought Y_i(X\delta\theta) i should has the same negative sign as ln(h(Y_i)), but A(X\theta) should have a positive sign. Please double check the issue and correct me if I am wrong. This will propagates to several derivations down the line.

3. One key contribution of this new interpretation is to show that the generalized case \mu \neq 1 is different. However, the manuscript does not provide any empirical or theoretical justification on the practical relevance of this case.

4. The writing of the paper can be improved, as currently there are way too many terms and pronouns appearing without clear definition or description. I want to bring one example here in the beginning of page 2:
"The consequence of this property of the average behavior is that our interpretation of the dropout-induced
regularization and optimization is ambiguous, when we think from the perspective of these decompositions.
That is, we know that these decompositions contain a scalar, which we are able to leave intact or attribute
to the fit and predict variables in accord with the rules of algebra"
The first sentence in the quoted message sounds like a repetition of the previous paragraph, which I don't really get the purpose. The second sentence then mentions a scalar which I don't really follow what it is. There are several events like this throughout the paper, such as when the term many-to-one first appears. It might be clear in authors head, and I can loosely figure it out as I read more, but this type of writing makes it a little difficult to follow the message clearly.

5. Following up on the writing, the paper currently has zero proposition, theorem, etc, which makes the main message difficult to gauge from a reviewer and reader perspective. Formalizing the derivation and conclusions can help improve the readability of the paper.

**Audience:**

Yes

**Audience Explanation:**

As a fundamental mechanism for machine learning, dropout and its interpretation should be of interest to the TMLR community.

**Claims And Evidence:**

No

**Claims Explanation:**

The manuscript’s core mathematical observation is mostly accurate in the settings where it is explicitly derived: if dropout appears inside a product, then the expected objective does not by itself remember whether the mask was interpreted as acting on the feature matrix or the parameter vector. However, I do view the claim that this gives a new perspective on the nature of dropout could be an overstatement, as the evidence are currently supporting linear regression, but much weaker for optimization behavior, neural network, and general dropout practices. Please refer to the weakness section, specifically point 1 to 3.

**Requested Changes:**

Please see weakness.

---

> ### Author Response · Authors · 2026-06-03
> **Reply to Review of Paper8789 by Reviewer ZBgj**
>
> We thank the reviewer for their thoughtful and detailed review. In what follows, we begin with how we use the term perspective, since this recurs in our work and reply. Then, we address the substantive points in turn.
>
> Perspective refers to the formulation of the decomposition, which we employ in order to interpret the regularization and optimization that we expect in practice. We use this term throughout the paper, i.e., the abstract, introduction, and Eq. 4, 5, 7, 9, 11, 16, 17, 18, 22, 23, 24, 25, 26, 33, 34, 35, 36, and 37.
>
> Our main contribution is twofold:
> - First, we identify the ambiguous nature of the decomposition in Eq. 4, 16, 22, and 33.
> - Second, we reformulate Eq. 4, 16, 22, and 33 based on fit and predict variables that model the canonical operations that a practitioner is able to perform in the learning process with dropout. These include the quantity to scale from train-to-test time and choice of whether or not to column normalize in the non-matrix factorization contexts.
>
> The significance of this is that we provide a practitioner with an interpretation at train time that corresponds to their train-to-test time workflow of interest.
>
> To address the question about the output at test time, we note in the penultimate paragraph of the introduction that the result in linear regression follows from the availability of a closed-form solution. Accordingly, we believe that our work provides a foundation for future work to either prove or disprove the equivalence in other contexts or establish empirical evidence to guide future proofs.
>
> ___
>
> In the case of generalized linear models, we wrote Eq. 28 such that the minus sign to the left of the summation distributes over the sum.
>
> ___
>
> In regard to $\mu \neq 1,$ we refer to the introduction of Helmbold and Long, which states: "Dropout in deep networks has a variety of other behaviors different from standard regularizers... In contrast, Wager et al. (2013) show that when dropout is applied to generalized linear models, the dropout penalty is always non-negative and does not depend on the labels." [1]
>
> - However, we found in all of the examined contexts that the penalty term under dropout depends on the data, parameters, and predictions at train time, when $\mu \neq 1$.
> - We note in the final paragraph of the introduction that the general case of the penalty term under dropout had not been considered.
> - Additionally, we explained in section 2.4.2 that the case of $\mu \neq 1$ requires at most a one-time cost of re-scale at test time in linear regression.
> - Indeed, the decision to scale the parameters and optimize over the re-scaled parameters implies that there is no cost of re-scale at test time. Also, we learn in this case that the local and global hyperparameters match, i.e., we observe that the square of the coefficient of variation of the dropout distribution appears throughout Eq. 9, 10, 11, and 12.
>
> In the other contexts, we are able to choose to scale the matrix factor, last layer weights, or parameters and optimize over the re-scaled quantity, so there is no need to re-scale at test time in accord with linear regression. Therefore, we view the case of $\mu = 1$ as a restrictive convention, since this particular value of the mean of the dropout distribution obscures the ambiguous nature of the decomposition, collapses terms in Eq. 13, 19, 27, and 38 that require future exploration, and reduces the square of the coefficient of variation to the variance of the dropout distribution.
>
> ___
>
> In regard to the clarity of the introduction, we note that the second paragraph explains that there is the ability to obtain a mathematical decomposition followed by the employment of that decomposition in the interpretation of the regularization and optimization. Subsequently, the fourth paragraph is about the many-to-one property of these decompositions, e.g., in linear regression the decomposition in Eq. 4 follows from $X\delta\theta = (X\delta)\theta = X(\delta\theta)$, $(X \odot \delta)\theta$, or $X (\delta \odot \theta)$, where the latter two products employ the Hadamard product with a dense dropout mask. Then, the fifth paragraph is about the consequence of this mathematical property, when we employ these decompositions in the interpretation of the regularization and optimization.
>
> To resolve the clarity around "scalar" in the fifth paragraph, we propose to revise to: "free scalar in the loss term".
>
> ___
>
> In regard to the format of the paper, we chose to take an exploratory approach with a case study of linear regression in the main body due to the familiarity of linear regression and the method of dropout in machine learning research. Then, we used this template to work through the other contexts in the appendix, and we overview the results from all of the contexts in the discussion and outlook section.
>
> ___
> ***References***
>
> [1] David Helmbold and Philip Long. Surprising properties of dropout in deep networks. Journal of Machine Learning Research, 2018.

---

### Review · Reviewer_W435 · 2026-05-21

**Summary Of Contributions:**

This paper studies the average behavior of dropout in four settings: linear regression, generalized linear models, matrix factorization, and fully connected networks with last layer dropout. The dropout mask is drawn from any distribution with finite non-zero mean and variance, generalizing the near universal assumption of Bernoulli or inverted dropout. The core observation is that the standard expected-loss decomposition is many-to-one in the dropped out quantity. The same decomposed form arises whether you drop the design matrix or the parameters, making the regularization strength and optimization objective ambiguous under purely algebraic reasoning.

The paper resolves this by introducing an operational perspective: enumerate the choices a practitioner actually makes (which quantity to scale, whether to column normalize), reformulate the decomposition in terms of those choices, and read off the corresponding regularization and optimization problem for each case. In linear regression (Section 2), all four reformulations produce identical test-time predictions. The invariant is the squared coefficient of variation of the dropout distribution. This closes an inconsistency in Srivastava et al. (2014), which produced two numerically incompatible regularization strengths for the same optimization problem. For matrix factorization, neural networks, and GLMs (Appendices B, C, D), the reformulations are derived but test-time equivalence is left open, signaled consistently by dashed lines in Figures 3, 5, and 7. The penalty-term analysis shows that for μ≠1, two additional prediction-dependent terms appear in the expected loss; these vanish under standard inverted dropout and have been absent from prior theory.

Strengths: the disambiguation resolves a genuine inconsistency across a substantial body of prior work; (σ/μ)^2 as the prediction-invariant is precise and new; the derivations are complete and self-contained; the paper is honest about what is proven versus open.

Weaknesses: "elementary operation" is never formally defined, leaving the enumeration's completeness unverified; the overlap with Clara et al. (2024) needs sharper delineation; Section 3.5's efficiency claim is stated without supporting analysis.

**Additional Comments:**

The solid/dashed line convention across Figures 1, 3, 5, and 7 is used honestly and is a good device. The derivations are self-contained; the appendices are well-organized. The component-wise diagonal mask (vs. the standard dense rectangular mask) is noted in Footnote 9 as simplifying the derivations; this is worth stating more prominently, since it explains why fewer equations are needed here than in prior work.

**Audience:**

Yes

**Audience Explanation:**

Dropout theory has been active for over a decade across NIPS, ICML, JMLR, and TPAMI. Resolving the concrete inconsistency in Srivastava et al. (2014), where the same optimization problem yields two incompatible regularization strengths (λ=p(1−p) vs. λ=(1−p)/p), is a result worth having on record. Identifying (σ/μ)^2 as the prediction-invariant tells practitioners that what matters is the ratio of variability to mean of the mask distribution, not variance or keep probability in isolation. The observation that PyTorch, TensorFlow, Keras, and MLX only implement μ=1 cases, locking out the richer μ≠1 penalty structure, is practically motivated and should prompt both theoretical and experimental follow up. Researchers working on regularization theory, implicit bias, or the dropout-Tikhonov connection will find this paper directly relevant.

**Broader Impact Concerns:**

None. Theory paper on the mathematics of dropout regularization. No ethical concerns.

**Claims And Evidence:**

Yes

**Claims Explanation:**

The mathematical claims are correct where proofs are provided. The many-to-one property (Eq. 4.. Eqs. 16, 22, 33 in the other contexts) follows directly from variance decomposition applied to the composite dropout product Xδθ. The test-time equivalence for linear regression (Eqs. 6, 8, 10, 12) is established in Appendix A.1.4 via a clean matrix identity: substituting A = XG^{-1} and a = G^{-1}x collapses both prediction forms. The penalty term decompositions (Eqs. 13, 19, 27, 38) are arithmetically correct. Cited results from Srivastava et al. (2014), Wager et al. (2013), Cavazza et al. (2018), Mianjy et al. (2018), and Helmbold and Long (2015, 2018) are accurately characterized.

Two issues require attention before I can recommend acceptance without reservation.

First, "elementary operation" is the load-bearing concept in the disambiguation framework but is never formally defined. Section 2.2 enumerates four choices for linear regression without stating what structural property makes an operation "elementary" or why the list is exhaustive. A practitioner might also post-scale fitted parameters, or normalize rows instead of columns; there is no argument that these fall outside the admissible class. The disambiguation claim is sound for the four enumerated cases but overreaches for the full space of practitioner choices. This matters more in the three non-linear-regression contexts, where test-time equivalence is already an open question and the set of distinct prediction behaviors is not yet fully characterized.

Second, Section 3.4 states that Clara et al. (2024) treated the abstract-distribution decomposition in their discussion section, without specifying which contexts that covers or what the present paper adds relative to it. TMLR's editorial policy requires that submissions not incorrectly claim novelty over prior published work. The paper should spell out: what exactly Clara et al. (2024)'s treatment states and for which setting, whether any results in Sections 2.1.1, 2.4, or the appendices are already contained there, and what is genuinely new here. If the overlap is minor (a paragraph of discussion vs. full derivations and reformulations), the paper should say so directly.

**Requested Changes:**

**Critical**

- Provide a formal definition of "elementary operation" and an argument that the four reformulations in each context are exhaustive within that class. Alternatively, reframe the disambiguation result as covering the specific enumerated choices, not the full space of practitioner decisions, and adjust the claims in the abstract and introduction accordingly. The current framing commits to more than what is proven, particularly for the three contexts where test-time equivalence is already open.

- In Section 3.4, make the overlap with Clara et al. (2024) explicit: state which contexts their abstract-distribution decomposition covers, whether any results in Sections 2.1.1, 2.4, or the appendices are already present there, and what is new in the present paper relative to that treatment. If the present paper provides the full derivation, reformulation, and operational interpretation that Clara et al. (2024) do not, state that directly.

**Strengthening**

- Section 3.5 claims the component-wise approach is "more sample and memory efficient" than the standard approach. A brief complexity comparison, e.g., O(n) or O(r) draws vs. O(mn) for the standard approach in linear regression and matrix factorization, would make this concrete and checkable.

- Footnote 5 identifies (σ/μ)^2 as the ridge parameter in a normal-normal Bayesian model with μ^2 scaling the prior and σ^2 the likelihood. This connects the dropout invariant to a prior-to-likelihood variance ratio, which is a genuine insight. It belongs in the main text as a short paragraph, not buried in a footnote.

- Sections B.4, C.4, and D.5 state that test-time equivalence is open and show a dashed arrow. A sentence on what additional structure would be sufficient to resolve the question, or whether there is reason to suspect non-equivalence, would make these sections useful as starting points for follow-on work rather than dead ends.

- Section 3.5 proposes transformer and diffusion experiments as future work but the proposal is disconnected from the paper's proven results. Tighten it to a few sentences or fold it into the discussion, making clear these are directions grounded in the theory rather than contributions of the present submission.

---

> ### Author Response · Authors · 2026-06-05
> **Reply to Review of Paper8789 by Reviewer W435**
>
> We thank the reviewer for their careful and perceptive review, which we believe captures the contributions and limitations of our work. In what follows, we begin with the point about the elementary operations. Then, we address the other points in turn.
>
> In particular, we plan to revise from elementary operations to the canonical operations that a practitioner is able to perform in the learning process with dropout. These include the quantity to scale from train-to-test time and choice of whether or not to column normalize in the non-matrix factorization contexts.
>
> In this way, we consider standard dropout and the converse quantity to scale, e.g., the parameters in section 2.2.2 in the case of linear regression. Then, the choice of column normalization in the non-matrix factorization contexts stems from the simplification of the regularization term, gradient, and Hessian that we observe throughout the work, e.g., the gradient and Hessian in sections D.4.2 and D.4.3 in the case of generalized linear models.
>
> Thus, we do not model every possible operation that a practitioner is able to apply in the learning process with dropout, and we leave open the consideration of other operations to future work. We plan to revise the abstract, introduction, and discussion and outlook sections in accord with this explanation.
> ___
> In regard to the work of Clara et al., we note this part of the introduction: "This motivates our approach to directly derive the statistical properties of gradient descent iterates with dropout. We study the linear regression model due to mathematical tractability and because the minimizer of the explicit regularizer is unique and admits a closed-form expression." [1] Then, the main body of the work is in terms of the Bernoulli decomposition, where the third section presents some statistical properties that extend the works of Srivastava and Srivastava et al. [2, 3]
>
> Additionally, the discussion and outlook section includes a subsection titled: "Generic dropout distributions", which states: "Analyzing a generic dropout distribution with mean $\mu$ and variance $\sigma$ may also paint a clearer picture of how the dropout noise interacts with the gradient descent dynamics." Then, the work expresses Eq. 4 and the closed-form solution when we optimize in terms of the original parameters. Notably, this is the solution in section 2.2.1 between Eq. 5 and Eq. 6, except the solution is expressed in terms of the original fit and predict variables.
>
> Thus, we view the results as a generalization of the works of Srivastava and Srivastava et al. However, the generalization never reached the problem of the ambiguous nature of the decomposition, reformulation in terms of operational fit and predict variables, equivalence of predictions at test-time, local vs. global hyperparameters, and the penalty term under dropout. Correspondingly, we plan to update the contextualization of prior works in section 3.4 with this material.
> ___
> In regard to the complexity comparison, we included the details in section 3.4 and referenced this with a footnote in section 3.5. To improve readability, we plan to turn the footnote into part of the first paragraph in section 3.5.
> ___
> Similar to the previous point, we plan to revise the footnote about the normal-normal Bayesian model and make this the final paragraph in section 2.2.1.
> ___
> In regard to the end of sections B.4, C.4, and D.5, we plan to revise the paragraphs to include a final sentence about approximations, non-uniqueness of a minimizer, and the establishment of empirical evidence to guide future proofs.
> ___
> With respect to section 3.5, we plan to revise the title to "Proposed experiments with a granular approach to dropout".
> ___
> ***References***
>
> [1] Gabriel Clara, Sophie Langer, and Johannes Schmidt-Hieber. Dropout regularization versus $\ell_{2}$-penalization in the linear model. Journal of Machine Learning Research, 2024.
>
> [2] Nitish Srivastava. Improving neural networks with dropout. University of Toronto, 2013.
>
> [3] Nitish Srivastava et al. Dropout: A simple way to prevent neural networks from overfitting. Journal of Machine Learning Research, 2014.

---

### Review · Reviewer_WuML · 2026-05-23

**Summary Of Contributions:**

The paper analyzes $\mathbb{E}[J(\theta)]$ under dropout across linear regression, matrix factorization, last-layer neural networks, and GLMs, generalizing the dropout distribution to any finite-mean ($\mu \neq 0$), finite-variance ($\sigma^2$) distribution. It claims the decomposition is "many-to-one" in the dropped-out quantity, creating ambiguous interpretations of regularization strength. Operational reformulations are proposed to resolve this. For linear regression, all reformulations yield identical predictions governed by $\left(\frac{\sigma}{\mu}\right)^2$. For $\mu \neq 1$, additional data- and prediction-dependent penalty terms emerge.

**Additional Comments:**

The paper applies correct mathematics to a misidentified problem. The genuinely novel observations, $\mu \neq 1$ penalty structure, $\left(\frac{\sigma}{\mu}\right)^2$ invariance, gap in library implementations, are buried under unnecessary formalism and unsupported by experiments. Restructuring around these narrower contributions, with empirical validation, would make for a stronger submission.

**Audience:**

Yes

**Audience Explanation:**

**Marginally.** The $\left(\frac{\sigma}{\mu}\right)^2$ invariance and $\mu \neq 1$ penalty analysis have some theoretical interest. The primary "ambiguity resolution" framing and lack of experiments severely limit appeal.

**Claims And Evidence:**

No

**Claims Explanation:**

- **The central contradiction is a change of variables, not an ambiguity.** The two "incompatible" regularization strengths (p.4) — $\lambda = p(1-p)$ penalizing $\theta$ vs. $\lambda = \frac{1-p}{p}$ penalizing $\tilde{\theta} = p\theta$ — produce the same total penalty: $\frac{1-p}{p}\|p\theta\|^2 = p(1-p)\|\theta\|^2$. The $\tilde{\theta}$ is not independently learned; it is algebraically derived from $\theta$. The entire resolution framework addresses a non-existent problem.

- **The many-to-one property is trivial.** $(X\delta)\theta = X(\delta\theta)$ is associativity of multiplication, not a finding about dropout.

- **Three of four settings are unresolved.** The prediction equivalence is proven only for linear regression. Matrix factorization (B.4), neural networks (C.4), and GLMs (D.5) are all left open.

- **No experiments whatsoever** — no numerical verification, no empirical investigation of open questions, no practical relevance demonstrated.

- **$\mu \neq 1$ penalty terms (Eq. 13) are identified but never motivated** — no evidence they help or hurt generalization.

**Requested Changes:**

- **Reframe the contradiction.** Acknowledge that $\lambda_1\|\theta\|^2 = \lambda_2\|\tilde{\theta}\|^2$ and clarify what genuine problem remains.
- **Add experiments:** (a) verify linear regression equivalence numerically, (b) test open equivalences empirically, (c) compare $\mu = 1$ vs. $\mu \neq 1$ on real tasks.
- **Resolve at least one open question** (B.4, C.4, or D.5) — analytically or empirically.
- Condense derivative appendices (B.3–D.4) — identical pattern repeated across ~8 pages.
- Formalize or remove "principle of machine learned interpretation" (p.2) — currently undefined.
- Label Section 3.5 proposed experiments as future work.

---

> ### Author Response · Authors · 2026-06-03
> **Reply to Review of Paper8789 by Reviewer WuML**
>
> We thank the reviewer for their thorough and candid assessment of our work. In what follows, we address the points in turn, where we begin with the point about the many-to-one property.
>
> To this end, we note that $X\delta\theta$ enables us to obtain Eq. 4 from Eq. 3. Instead, if we employ the Hadamard product, then we are able to obtain Eq. 4 in two different ways, where we do not use the associative property of $X\delta\theta$.
> - Namely, we begin with the standard approach, where we dropout the design matrix with a dense dropout mask $\delta$, i.e., $(X \odot \delta)\theta$.
> - Then, we dropout the parameters with a dense dropout mask $\delta$, i.e., $X (\delta \odot \theta)$.
>
> Subsequently, if we employ Eq. 4, then we have a free scalar $\mu$ in the loss term, which we are able to leave intact or attribute to the fit and predict variables. But Eq. 4 does not suggest a way to do this in accord with the original dropped-out quantity or implementation of interest. This is the problem of the ambiguous nature of the decomposition.
> ___
> In regard to the multiple interpretations, we recall that the works considered the standard approach and presented an interpretation, where the regularization strength is written in terms of the variance. [1, 2] Then, the works provided a different interpretation in the final paragraph: "Another interesting way to look at this objective is to absorb the factor of $p$ into $w$", where $w$ becomes $\theta$ in our notation. "This makes the dependence of the regularization constant on $p$ explicit. For $p$ close to $1$, all the inputs are retained and the regularization constant is small. As more dropout is done (by decreasing $p$), the regularization constant grows larger."
>
> Now, if we consider the perspective of a practitioner, then the works provide two different interpretations of the regularization strength in the same optimization problem. We claim that the problem is not in the equivalence of penalty or closed-form solution but in the way that these interpretations teach a practitioner to reason about the choice of $p$.
>
> In contrast, we provide a practitioner with an interpretation at train time that corresponds to their train-to-test time workflow of interest. Then, we note in the penultimate paragraph of the introduction that the result in linear regression follows from the availability of a closed-form solution. Thus, we believe that our work provides a foundation for future work to either prove or disprove the equivalence in other contexts or establish empirical evidence to guide future proofs.
> ___
> In regard to $\mu \neq 1,$ we refer to the introduction of Helmbold and Long, which states: "Dropout in deep networks has a variety of other behaviors different from standard regularizers... In contrast, Wager et al. (2013) show that when dropout is applied to generalized linear models, the dropout penalty is always non-negative and does not depend on the labels." [3]
> - However, we found in all of the examined contexts that the penalty term under dropout depends on the data, parameters, and predictions at train time, when $\mu \neq 1$.
> - Additionally, we explained in section 2.4.2 that the decision to scale the parameters and optimize over the re-scaled parameters implies that there is no cost of re-scale at test time.
>
> In the other contexts, we are able to choose to scale the matrix factor, last layer weights, or parameters and optimize over the re-scaled quantity, so there is no need to re-scale at test time. Therefore, we view the case of $\mu = 1$ as a restrictive convention, since this particular value of the mean of the dropout distribution obscures the ambiguous nature of the decomposition, collapses terms in Eq. 13, 19, 27, and 38 that require future exploration, and reduces the square of the coefficient of variation to the variance of the dropout distribution.
> ___
> In regard to the computations, we prefer to keep these intact to enable the interdisciplinary audience of the journal to consult each context of interest as a self-contained reference.
> ___
> In regard to the "principle of machine learned interpretation", the motivation behind the terminology stems from the earlier point about the ambiguous nature of the decomposition in Eq. 4, 16, 22, and 33. That is, we require a principle that goes beyond the rules of algebra and constrains the way that we are able to manipulate the free scalar in the loss term of these decompositions.
> ___
> In regard to the title of section 3.5, we plan to revise the title to "Proposed experiments with a granular approach to dropout".
> ___
> ***References***
>
> [1] Nitish Srivastava. Improving neural networks with dropout. University of Toronto, 2013.
>
> [2] Nitish Srivastava et al. Dropout: A simple way to prevent neural networks from overfitting. Journal of Machine Learning Research, 2014.
>
> [3] David Helmbold and Philip Long. Surprising properties of dropout in deep networks. Journal of Machine Learning Research, 2018.

---

> > ### Comment · Reviewer_WuML · 2026-06-11
> > **Response**
> >
> > Thank you for taking the time to respond. I understand your focus, but I believe the interpretive inconsistency in prior literature (from 2013) is being given disproportionate weight and is not mathematically valid. Ultimately, the absence of empirical experiments leaves the paper's broader claims unsupported, and I remain unconvinced.

---

### Decision · Action_Editor_8LDi · 2026-06-23

**Recommendation:** Reject

**Additional Comments:**

This submission received two "leaning reject" and one "leaning accept" from the reviewers. Two reviewers gave "leaning reject" mainly for the reasons I summarized in the comment for **Are the claims made in the submission supported by accurate, convincing and clear evidence?**

However, all the reviewers find some of the contributions, such as the invariance of $(\sigma/\nu)^2$ and addressing the misbelief of dropout behavior in the unexplored case $\mu \neq 1$, novel and interesting. According to the reviewers' comments, reorganizing the paper to focus on these points could make the overall contribution clearer. Demonstrating their practical relevance through numerical experiments in setups not covered by the theory would also enhance the paper. See, in particular, Additional Comments of Reviewer WuML's review on these points.

**Audience:**

Yes

**Audience Explanation:**

Two reviewers marked "yes". One of them expressed that dropout and its interpretation are of interest to the TMLR community. The other reviewer mentioned that this work addresses "genuine inconsistency across a substantial body of prior work" on dropout. The last reviewer initially wrote marginally "yes" for the $(\sigma/\mu)^2$ invariance and $\mu \neq 1$ analysis but "no" in their final recommendation.

**Claims And Evidence:**

No

**Claims Explanation:**

After the authors' rebuttal, two reviewers expressed that some claims are still not sufficiently supported by convincing evidence.

One of them mentioned that the "incompatibility" of regularization interpretation of dropout is not a valid claim and this work addresses a misidentified problem, by showing that two equivalent parametrizations are possible through change of variables. They believe that numerical verifications/investigations are necessary to demonstrate practical relevance of the claims.

The other reviewer similarly wrote and that some of their questions are unanswered. More specifically, the current manuscript gives an impression that it provides a new perspective on the nature of dropout, although the evidence is limited to linear regression. They find the invariance of $(\sigma/\mu)^2$ and the issue with the common practice of setting $\mu = 1$ interesting but the practical relevance unclear from the manuscript.

The remaining reviewer is convinced by the claims and mentions that the mathematical claims provided with proofs are correct.

**Resubmission Of Major Revision:**

The authors may consider submitting a major revision at a later time.